# The Grueneberg ganglion controls odor-driven food choices in mice under threat

Julien Brechbühl [1,2], Aurélie de Vallière[1,2,3], Dean Wood[1,2,3], Monique Nenniger Tosato[1] &
Marie-Christine Broillet [1,2 ✉]

The ability to efficiently search for food is fundamental for animal survival. Olfactory messages are used to find food while being aware of the impending risk of predation. How these different olfactory clues are combined to optimize decision-making concerning food selection remains elusive. Here, we find that chemical danger cues drive the food selection in mice via the activation of a specific olfactory subsystem, the Grueneberg ganglion (GG). We show that a functional GG is required to decipher the threatening quality of an unfamiliar food. We also find that the increase in corticosterone, which is GG-dependent, enhances safe food preference acquired during social transmission. Moreover, we demonstrate that memory retrieval for food preference can be extinguished by activation of the GG circuitry. Our findings reveal a key function played by the GG in controlling contextual food responses and illustrate how mammalian organisms integrate environmental chemical stress to optimize decision-making.

[1] Department of Pharmacology and Toxicology, Faculty of Biology and Medicine, University of Lausanne, Bugnon 27, CH-1011 Lausanne, Switzerland.
[2] Department of Biomedical Sciences, Faculty of Biology and Medicine, University of Lausanne, Bugnon 27, CH-1011 Lausanne, Switzerland. [3]These authors contributed equally: Aurélie de Vallière, Dean Wood. ✉email: mbroille@unil.ch

Animals continuously assess the environmental risks and, for survival, may need to compromise between the gain of feeding and the potential cost of exposure to danger[1]. Especially, when confronted to food scarceness, behavioral trade-offs need to be made[2,3]. Olfactory senses play an important role in this process as they help to find food but also to sense and avoid threatening situations[4–9] such as predation. For rodents, the attraction to food is primarily determined by familiarity, as scents of known food are innately favored[2,10]. These scents are detected by olfactory sensory neurons (OSNs) found in the main olfactory epithelium (MOE)[11,12]. Food familiarity can be acquired via communication between individuals, by the so-called "social transmission of food preference" (STFP), a behavioral process that occurs by sniffing the breath of conspecifics and making an association between the novel food scent and the presence of the endogenously produced carbon disulfide gas ($CS_2$)[4,13,14]. For this process to occur, the concomitant recognition of the food odorant by OSNs and the detection of the $CS_2$ by guanylyl cyclase-D neurons (GC-D) of the MOE is essential. On the other hand, odors that signal an impending danger are involuntarily left behind by predators[15,16] or emitted by stressed congeners as warning alarm pheromones[9,17]. We showed previously that these danger and warning cues are detected in a specialized olfactory subsystem, the Grueneberg ganglion[9,15] (GG). They generate, in the recipient mouse, stereotypical fear-related behaviors, such as freezing and risk assessment as well as stress-related systemic responses[15,18].

In this study, we provide evidence and new biological insights on how mice exploit threatening scents to optimize and control their food selection. Using gene-targeted, surgically treated and sham-operated mice as well as a series of integrative behavioral assays, we functionally determine that the GG olfactory sub-system controls innate and acquired food preferences when mice smell an impending danger. The GG is indeed required to decipher the threatening quality of an unfamiliar food. We also find that its activation increases the systemic corticosterone level which in turn enhances safe food preferences acquired during social transmission. Moreover, we finally demonstrate that the acquisition of a food preference itself could be circumstantially reset by the activation of the GG circuitry, providing a decision-making advantage for mice.

## Results

### Selective deletion of the GG circuitry.
The conflicting environmental olfactory messages carried by odorants emitted by familiar food and by olfactory danger cues need to be continuously evaluated for risk taking and final food decision-making. To address the functional relevance of the GG in this last process in mice, we first surgically disconnected it by nerve axotomy[9] (axo; Fig. 1a). Thanks to the expression of the olfactory marker protein (OMP; Fig. 1b, c), a neuronal marker for both the GG and the MOE, we then verified that the axotomy induced specific loss of the GG in axotomized (Axo) but not in control sham-operated mice (Ctrl) (Fig. 1b) without affecting the MOE (Fig. 1c). We next demonstrated that the GC-D neurons were still present in the MOE after GG axotomy (Fig. 1c) thanks to the expression of the enzyme phosphodiesterase 2 A (PDE2A; Fig. 1b and c) which is shared by GG and GC-D circuities[19]. We then used transgenic guanylyl cyclase-G (GCG, a marker of GG circuitry)–Cre–green fluorescent protein (GFP) mice[19] to selectively trace the GG connections into their olfactory bulb target (OB; Fig. 1a), the necklace glomerular complex (NG; Fig. 1a). Thereby, we distinctly differentiated, in the NG, the parallel connections between GG and GC-D circuitries and we found that the GC-D circuitry and, in particular its associated necklace glomeruli (NG), were still intact in GCG–Cre–GFP Axo mice (Fig. 1d and Supplementary Fig. 1; Ctrl, $N_{mouse} = 3$; Axo, $N_{mouse} = 5$). Moreover, this GC-D circuitry was still active in the presence of $CS_2$, independently from the GG axotomy procedure, as verified by immediate-early gene c-Fos expression[15] (Fig. 1e, f; Ctrl, $N_{glomeruli} = 6$; Axo, $N_{glomeruli} = 7$; two-tailed $t$-test: p = 0.544, ns). We have then systematically used the selective deletion of this olfactory subsystem to functionally dissect the relevance of the GG circuitry in odor-driven food selection when mice were exposed to chemical danger cues.

### GG encodes threatening scents and dominates over MOE-transmitted odor signals.
Mice naturally disregard unfamiliar food based on the unknown odorants it releases, moreover familiar food soiled with danger cues is likewise avoided[2]. As a first physiological assay, we, therefore, tested odorants commonly used to odorize food such as non-synthetic spices[13] and we found that these odorants were not directly detected by GG neurons. Indeed, we revealed with calcium imaging experiments performed on acute Fura-2 acetoxymethyl ester (Fura; Fig. 2a) loaded GG slice preparations from transgenic OMP-GFP mice[9] that, in a total of 54 living GFP tagged GG neurons ($N_{mouse} = 4$; $n_{slice} = 6$), Cinnamon, Cocoa, Anise, Oregano, Thyme, Basil, Nutmeg, Ginger as well as the standard mice food and $CS_2$ did not generate any GG neuronal activity (Fig. 2b). On the other hand, the predator-derived cues 2-propylthietane (2PT) from the stoat anal glands, the mouse alarm pheromone 2-sec-butyl-4,5-dihydrothiazole (SBT) as well as mountain lion (Mt.Lion) urine[15,20] initiated reversible calcium responses in respectively 83, 57 and 100% of the GG neurons. Thus, this first approach not only suggests that the unfamiliarity of a food encoded by its emitted odorants could not be directly deciphered by the GG but also confirmed the ability of the GG to identify danger cues potentially emitted by soiled food.

We next verified, in an integrative context of food choice, the implication of the GG in decoding food unfamiliarity based on the odorants emitted. Ctrl and Axo mice were challenged to select, between familiar and unfamiliar food in a two choices assay (Fig. 2c and Supplementary Fig. 2a). To that purpose, two familiar powdered foods were proposed to mice, odorized either with a series of never-encountered before odorants (odor #1; as unfamiliar food) or with the odorless water (odor #2; as familiar food). We placed all the tested odorants around the powdered foods to exclude any toxicity potentially displayed by synthetic cues. Subsequently, the preference for an odorized food was calculated as the ratio between the consumption of food odor #1 versus the total food consumed (food odor #1 + odor #2) in which the 0.5 value, corresponding to the non-preference threshold was subtracted. Values were thus expressed between 0.5 and −0.5 where positive scores corresponded to a preference for the food odor #1; negative scores for the food odor #2 and zero corresponded to no preference displayed. Hence we showed that rodents indeed prefer familiar food[2] as non-synthetic spices such as the unfamiliar Cinnamon or Cocoa[13] as odor #1 were not preferred by mice. Moreover, we observed that this innate choice was also performed by Axo mice (Fig. 2d), confirming that this avoidance behavior was indeed not directly dependent of the GG detection (Fig. 2a, b) but also pointed out to the conserved MOE functionality in the GG Axo mouse model. Then we found that the GC-D-related ligand $CS_2$, without any social context[10], did not influence diet selection in both Ctrl and Axo mice (Fig. 2d). We next tested butyric acid (BA) a known aversive odorant that smells rancid and has no alerting relevance[15] and found that it indeed generated food avoidance both in Ctrl and Axo mice confirming its previously reported GG-independent detection[15]

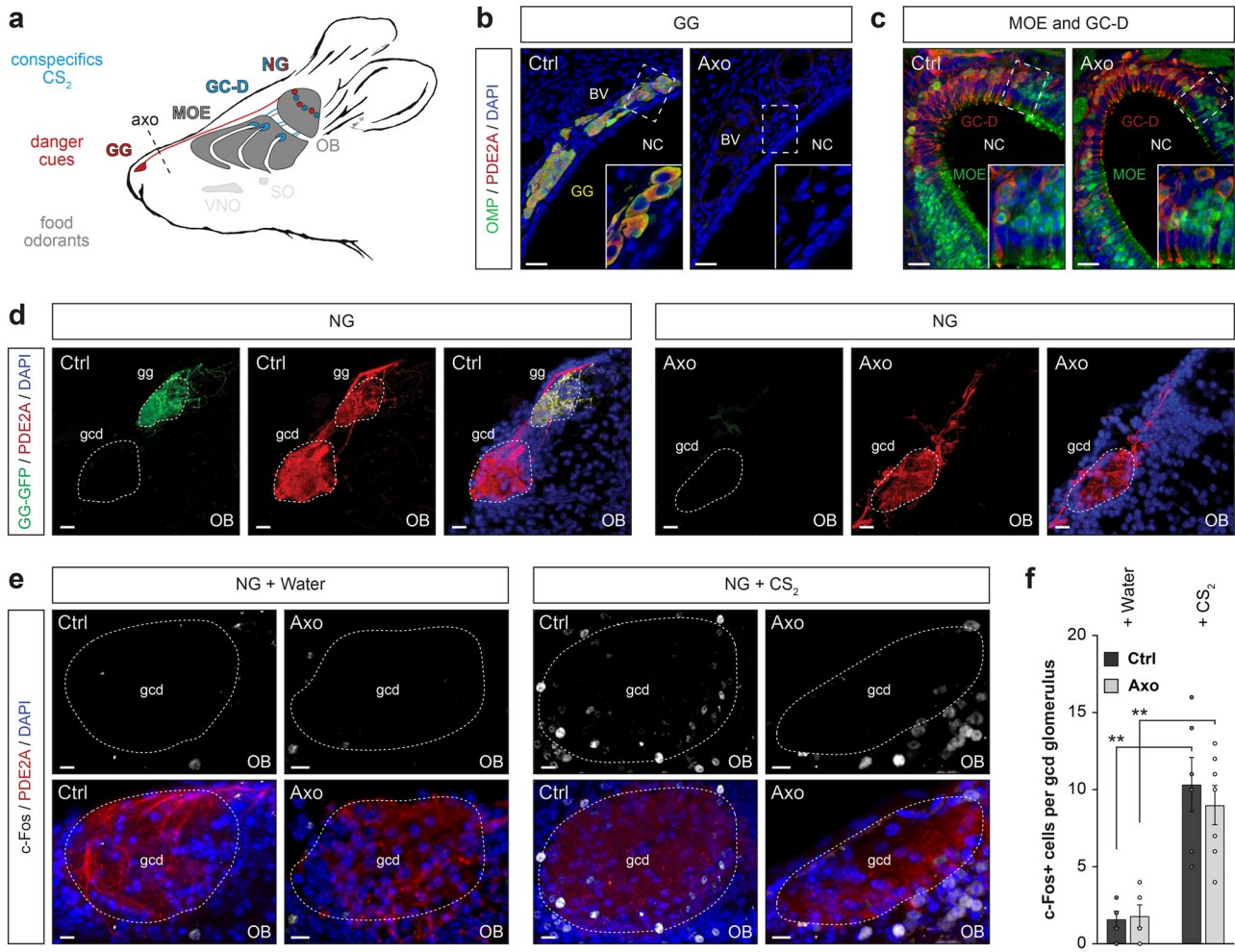

**Fig. 1 Selective GG olfactory subsystem deletion by axotomy. a** Schematic representation of a mouse head and its olfactory subsystems. GG: Grueneberg ganglion (red); GC-D: guanylyl cyclase type D (blue); MOE: main olfactory epithelium; SO; septal organ; VNO: vomeronasal organ; NG: necklace glomeruli (red and blue); OB: olfactory bulb. Localization of the GG axotomy procedure is represented by a black dashed line (axo). **b, c** Maximum-intensity projection of a double immunostaining (anti-OMP, OMP, in green; anti-PDE2A, PDE2A, in red; colocalization, in yellow) of the GG (**b**) and MOE/GC-D (**c**) region in Ctrl and Axo mice. Slice views of detailed regions of dashed white rectangles highlight the presence of MOE (OMP + | PDE2A-) and GC-D (OMP- | PDE2A+) but absence of GG (OMP + | PDE2A+) neurons in Axo mice. NC: nasal cavity; BV: blood vessel. **d** GG axotomy leads to the deletion of gg glomeruli without affecting the gcd glomeruli. **e** Representative c-Fos (anti-c-Fos, c-Fos, in white) activity of GC-D glomeruli observed in Ctrl and Axo mice after conditioning either with water (NG + Water, basal activity) or with 10 ppm of $CS_2$ (NG + $CS_2$, stimulated activity). **f** Quantification of c-Fos activity observed in GC-D glomeruli for Ctrl (black) and Axo (gray) mice. Data are represented as mean ± SEM with aligned dot plots for individual mice values. For comparisons between conditions and between Ctrl *vs.* Axo mice, two-tailed Student's *t*-tests or Wilcoxon *w*-tests are used, **$p < 0.01$. A minimum of 5 glomeruli were used per condition. Scale bars are 25 µm (**b, c**), 20 µm (**d**) and 10 µm (**e**). DAPI counterstains are shown in blue. Glomeruli are delimited with dashed white lines.

(Fig. 2d). Food odorized with predator scents were also aversive (Fig. 2d) as observed with the pyrazine analogues (Pyrazines), found for example in Mt.Lion urine[20], 2-PT, 2,4,5-trimethylthiazoline (TMT) from the red fox feces as well as the mouse alarm pheromone SBT. Nevertheless, in Axo mice, this dislike was reduced indicating and confirming (Fig. 2b) that the GG was indeed implicated in the perception of these chemical danger cues[15]. Finally, we used Mt.Lion urine as a natural source of predator scents[20] and found that its aversive effect on food choice was exclusively dependent on a functional GG (Fig. 2d). Thus, confirming our GG calcium imaging investigations (Fig. 2a, b) that show that mice do not recognize odor signals emitted by unfamiliar food via GG detection as demonstrated by exposing them to familiar versus unfamiliar foods (Fig. 2d).

We next highlighted that this olfactory subsystem was fundamental for mice to decipher the threatening quality of an unfamiliar food (Fig. 2e). Indeed, on sets of naive mice, when we used the pungent and unfamiliar BA as odor #2 (Fig. 2e) in the previous two choices assay (Fig. 2c), Ctrl and Axo mice now preferred Cinnamon, Cocoa and $CS_2$ as sources of unfamiliar odorized food (Fig. 2e) confirming the previously observed aversity of the BA (Fig. 2c). Interestingly, in the presence of danger cues such as the Pyrazines, 2PT, TMT, SBT and Mt.Lion urine, Ctrl mice now systematically preferred the aversive and unfamiliar BA (Fig. 2e). TMT and SBT were particularly efficient as they were still able to generate this innate reaction in serial dilutions (Supplementary Fig. 2b, c). Remarkably, this observed preference for BA disappeared in the absence of a functional GG, without affecting the total food consumption (Supplementary Fig. 2d, e). Thus, mice decode the threatening quality of unfamiliar food by GG detection.

Taken together, our results show that, the GG acts as an immediate sensor, which deciphers the threatening quality of

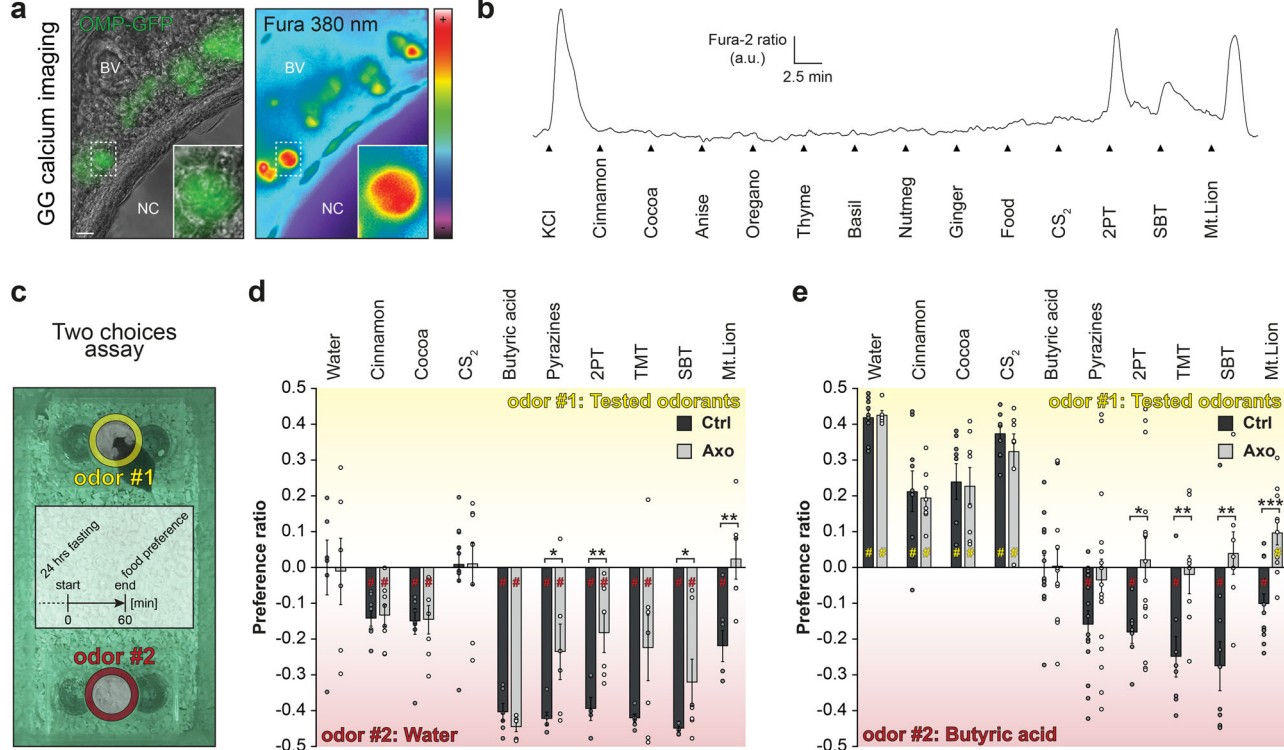

**Fig. 2 A functional GG is required to decipher the threatening quality of an unfamiliar food. a** Representative GG calcium imaging from an OMP-GFP mouse loaded with Fura-2AM (Fura), observed here at 380 nm in color-encoded map for unbound Fura. NC: nasal cavity; BV: blood vessel. Scale bar, 20 μm. **b** Representative continuous recording of a responding GFP+ GG neuron (dashed white rectangle in (**a**)) performed with Fura-2 ratio (340 nm/380 nm, in arbitrary units a.u.). Control pulse of KCl (25 mM) determines the cellular viability. Tested solutions are listed; Spices and Food (1:100), CS$_2$ (10 ppm), 2PT and SBT (1:5000), Mt.Lion (1:500). **c** Two choices assay, illustrated here with an infrared snapshot and a procedure time-table. Odorants were placed around food resources (odor #1, yellow; odor #2, red). **d, e** Quantification of food Preference ratio for Ctrl (black) and Axo (gray) mice, calculated as the ratio of food consumption [(odor #1)/(odor #1 + odor #2)] - 0.5. Positive scores display a preference for the food odor #1; negative scores for the food odor #2. Tested odor #1 (10%, or pure for Mt.Lion) are indicated and are opposed to odor #2 (Water, in (**d**) and BA (10 %), in (**e**)). Data are represented as mean ± SEM with aligned dot plots for individual mice values. Calculation of statistical significances of the preference ratio is performed with Z tests, #$p < 0.05$ (yellow or red # for a preference respectively for odor #1 or odor #2); non-significant if not mentioned. For comparisons between Ctrl vs. Axo mice, two-tailed Student's t-tests or Wilcoxon w-tests are used, *$p < 0.05$; **$p < 0.01$. 6 to 16 animals are used per condition.

odorants emitted by food. Mice will take their final consumption decision about the safety of the resource thanks to its smell and to its GG perception.

**Threatening scents activate GG-dependent corticosterone responses.** In the wild, food resources are often limited and could be located in a dangerous setting such as impending predation. Nevertheless, the motivational state is modified under hunger context[3]. To evaluate trade-offs displayed by fasting mice when confronted to actively searching a food resource in a danger context, we next challenged Ctrl and Axo mice with unreachable food resources moistened with SBT and with Mt.Lion urine, as sources of respectively intra- and inter-species conditioning stimuli (+C.S) mimicking environmental evidence of olfactory threats (Fig. 3a). Remarkably, we observed the absence of a fear-like response such as freezing (Fig. 3b) and the display of a risk assessment behavior (Fig. 3c) in both Ctrl and Axo mice[15], indicating that, in a context of food scarceness, the opportunity to eat indeed overrules fear (Fig. 3a–c). These observed behavioral adaptations also highlight that odorants emitted by food were sufficient to initiate this motivational-related behavioral process independently from GG detection (Fig. 3c). We then followed the systemic integration of these stressful situations with the expression of the immediate-early gene c-Fos[21]. We focused on the amygdalopiriform transition area (APir; Fig. 3d) which is the

brain region implicated in the increase of stress-related hormone level in the blood when mice smell volatile predator scents[15,22]. We found that this brain region was significantly activated by both SBT and the predator urine (Fig. 3e, f). We next confirmed this result by measuring an increase of the systemic corticosterone level (Fig. 3g), implying that intra- and inter-species danger cues were both processed in this specific APir nucleus. In a dilution series of intraperitoneal corticosterone injections (Cort. i.p.; Fig. 3h), we were further able to mimic the observed stress-related hormonal elevation with an amount of Cort. i.p. of 5.0 mg kg$^{-1}$, therefore bypassing APir activation in absence of conditioning stimuli (−C.S; Fig. 3h and Supplementary Fig. 3). Moreover, we found that, in Axo mice, the activation of the APir region, as well as the elevation of the systemic corticosterone, were significantly impaired in this threatening context (Fig. 3e–g), demonstrating that, although the GG was not involved in odor-driven foraging, it was essential for the hormonal and physiological adaptation to danger sensing.

**Elevation of corticosterone optimizes learned-food selection.** In the search of food resources, knowledge of familiar food obtained by STFP support the food decision-making by a recall memory process that requires the activation of a specific part of the hippocampus, the ventral subiculum[13,23] (VS; Fig. 3d). In this study, we found that the presence of danger cues enhanced this learned

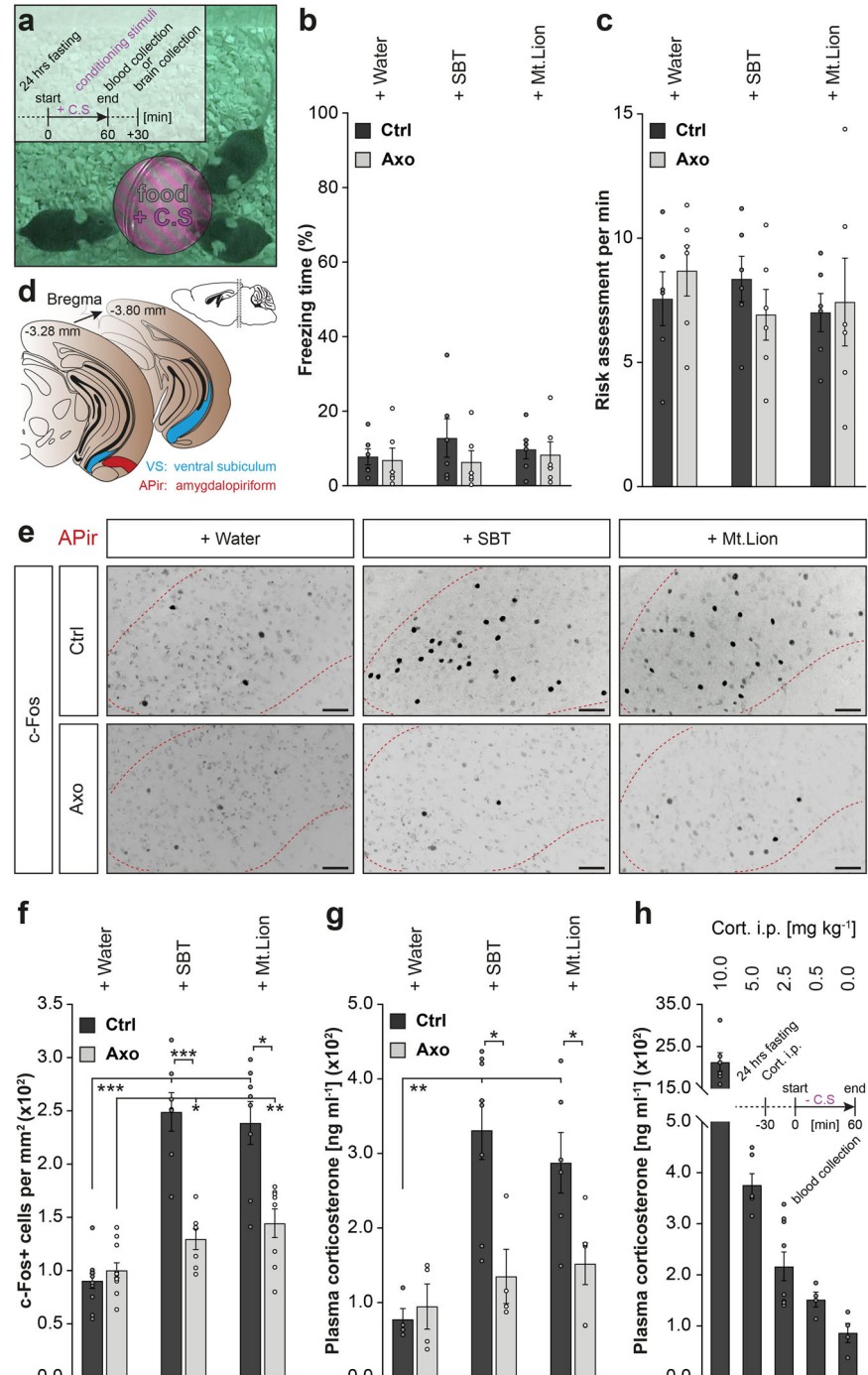

**Fig. 3 Detection of chemical danger cues by the GG drives systemic corticosterone elevation via APir activation. a** Infrared snapshot and procedure time-table illustrating mice in contact with an unreachable food resource (food, gray), moistened with conditioning stimuli (+C.S, purple). Here, Ctrl mice displayed a typical risk assessment behavior in the presence of + Mt.Lion. **b-c** Quantification of the absence of freezing response expressed as a percentage of time (**b**) and of the risk assessment mean occurrence per minute (**c**) displayed by mice during the behavioral assay (**a**). **d** Serial coronal brain sections corresponding to Bregma (−3.28 to −3.80) used for the c-Fos investigation of the amygdalopiriform (APir, red) and the ventral subiculum (VS, blue) areas. **e** Representative c-Fos stainings (dark spots) in Ctrl and Axo mice in APir (dashed red lines) obtained after + Water, + SBT (1:500) or + Mt. Lion (pure urine) conditioning. Scale bars, 50 μm. **f** Quantification of the density of c-Fos+ cells observed in APir. **g** Plasma corticosterone analysis after + C.S stimulation. **h** Procedure time-table illustrating the plasma corticosterone analysis in absence of conditioning stimulus (− C.S) and after intraperitoneally injection of corticosterone (Cort. i.p., 10.0, 5.0, 2.5, 0.5, 0.0 mg kg⁻¹). (**b**, **c** and **f-h**) Values obtained from Ctrl (black) and Axo (gray) mice are represented as mean ± SEM with aligned dot plots. For comparisons between conditions and between Ctrl *vs.* Axo mice, two-tailed Student's *t*-tests or Wilcoxon *w*-tests are used, *$p < 0.05$; **$p < 0.01$; ***$p < 0.001$. A minimum of four animals were used per condition.

food-odorant preference when we performed a two choices assay under threatening conditions. For that, we first trained mice to develop a food-odorant preference. In brief, a demonstrator mouse ate an odorized demonstrating food (de.food; standard powdered food odorized for example with Cinnamon as a spice #1; Phase 1; Fig. 4a). Then it returned with its observer littermates

to allow the STFP learning process to happen (Phase 2; Fig. 4a). Finally, after 24 h of fasting, the observer mice were individually confronted to a two choices assay between two odorized foods, the demonstrating food and a novel food (new.food; standard powdered food odorized for example with Cocoa as spice #2; Phase 3; Fig. 4b), both surrounded with the same conditioning

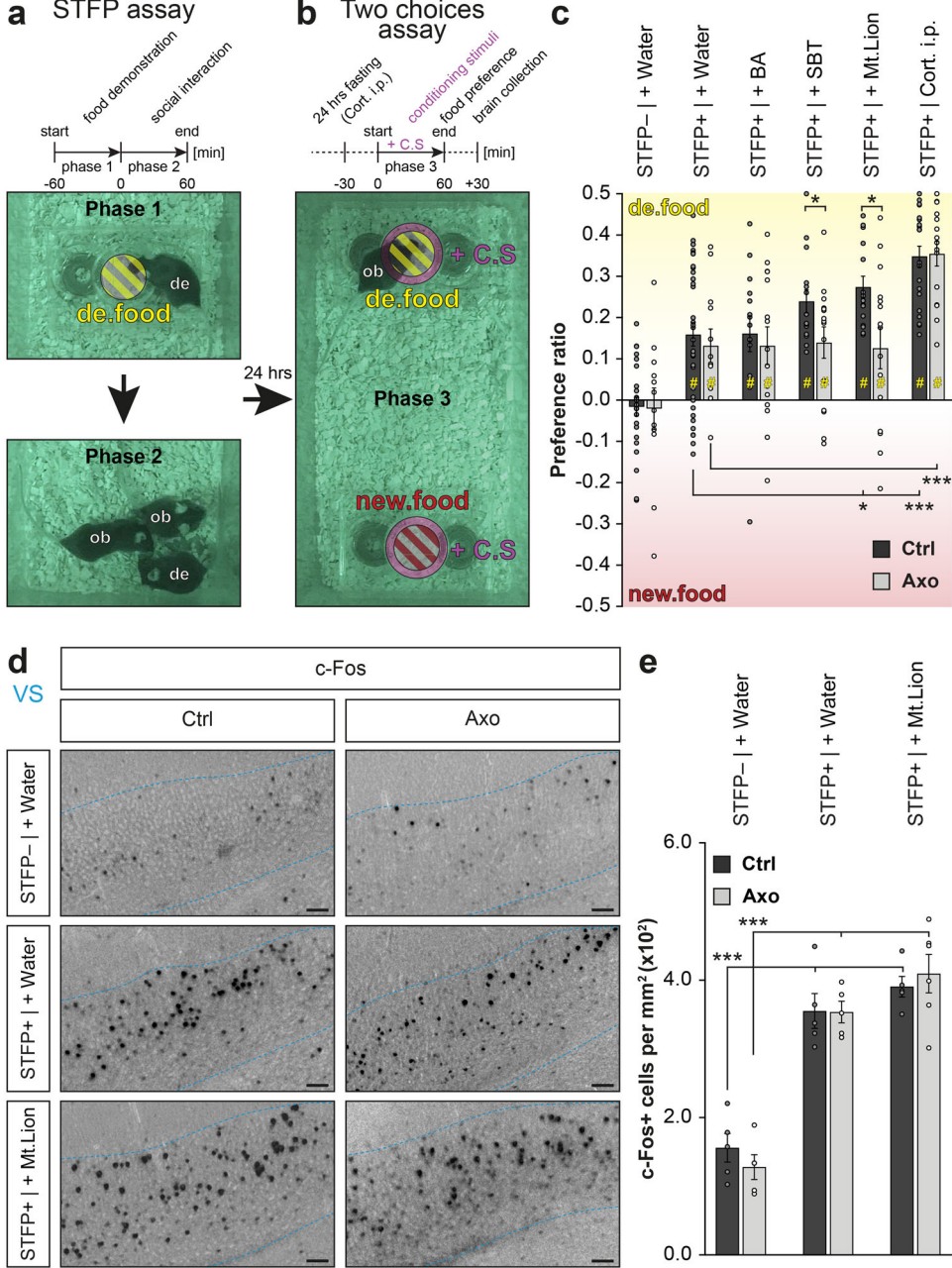

**Fig. 4 Chemical danger cues enhance food preferences acquired by STFP. a** Acquisition of a food preference performed by STFP assay is illustrated by infrared snapshots and a procedure time-table. In phase 1, a demonstrating food (de.food, yellow) is presented to a demonstrator mouse (*de*). In phase 2, observer mice (*ob*) are in contact with the *de* for social interactions and acquisition of a preference for the de.food. **b** In phase 3, each *ob* mouse is individually tested in a two choices assay with two sources of food (de.food, yellow; new.food, red) surrounded with the same conditioning stimulus (+C.S, purple). **c** Quantification of the food preference ratio in Ctrl and Axo mice without or with STFP procedure and under the indicated environmental conditioning (STFP− or + | + C.S) or after intraperitoneal injection of 5.0 mg kg$^{-1}$ corticosterone (Cort. i.p.). Statistical significances of preference ratio are performed with $Z$ tests, #$p < 0.05$ (yellow, for a preference for de.food); non-significant if not mentioned. 12 to 36 animals were used per condition. The following pairs of spices were used: cinnamon 1% *vs.* cocoa 2%; anise 1% *vs.* oregano 2.4%; thyme 2% *vs.* basil 1.4%. **d** Representative c-Fos staining (dark spots) in Ctrl and Axo mice under the indicated conditioning in VS (blue dashed line). Scale bars, 50 µm. The following pair of spices were used: nutmeg 1% *vs.* ginger 1%. **e** Quantification of the density of c-Fos+ cells in VS from (**d**). A minimum of 4 animals were used per condition. **c**, **e** Values obtained from Ctrl (black) and Axo (gray) mice are represented as mean ± SEM with aligned dot plots. For comparisons between conditions and between Ctrl *vs.* Axo mice, two-tailed Student's *t*-tests or Wilcoxon *w*-tests are used, *$p < 0.05$; ***$p < 0.001$.

stimulus (+C.S; Fig. 4b). To avoid any individual innate preference, spices were used in a counterbalanced mode[13] (Supplementary Fig. 4a to c). We observed in both Ctrl and Axo mice that STFP was, indeed, required to develop a food preference (STFP+|+ Water; Fig. 4c) that was, remarkably, not impacted by a chemical confusion generated by the pungent BA (STFP+|+ BA; Fig. 4c). These first effects not only confirmed the conserved functionality of the GC-D circuitry in Axo mice (Fig. 1c, e, f) but also pointed out that, under aversive distraction, mice still olfactively decipher food preferences. Surprisingly, we obtained, with the danger cue SBT (STFP+|+ SBT) and with Mt.Lion urine (STFP+|+ Mt.Lion) an enhancement of the food-odorant preference (Fig. 4c). Indeed, the observed mean food preferences were amplified by around 60 % compared to STFP under standard environmental (STFP+|+ Water) or under pungent conditions (STFP+|+ BA). This enhancement was observed in Ctrl but not in Axo mice. It was found to be both independent from food consumption (Supplementary Fig. 5a) and from an improvement of the recall memory process as no significant increase in c-Fos activity was observed in the VS brain area under these conditions (Fig. 4d, e). Interestingly, we found that injection of Cort. i.p. $5.0\,mg\,kg^{-1}$ under neutral context (STFP+|Cort. i.p.) was sufficient to mimic this behavioral improvement both in Ctrl and Axo mice (Fig. 4c) without affecting the memory-retrieval activity of the VS (Supplementary Fig. 5b, c). We thus demonstrated here that, when danger cues are sensed by the GG, the increase in the systemic corticosterone level that occurs (Fig. 3e) leads to an enhancement of the food preference previously acquired by STFP.

**GG circuitry activation selectively erases safety food memory**. Conspecific interactions are useful for mice to get food familiarity. We found here that mice can also benefit from threatening chemical information present in their environment to reset this previously acquired food-odorant familiarity when associated with a danger context. We indeed tested the impact of olfactory threats on the acquisition of a new food-odorant preference as we challenged mice 1 h after an STFP procedure (Phase 1; Fig. 5a) with a surrogate display that contained the unreachable demonstrating food moistened with an associated conditioning stimulus (Phase 2; +C.S; Fig. 5a), a procedure that allows food investigation (Fig. 3a–c). After 24 h of fasting, the observer mice were then tested in a two choices assay (Phase 3; Fig. 5b). Unexpectedly, we found that in association with the danger cues SBT (STFP+/+ SBT) or with Mt.Lion urine (STFP+/+ Mt.Lion), Ctrl mice did not display food-odorant preferences while they were still observed in Axo mice (Fig. 5c). As a control, we verified that GG-related ligands directly associated with a demonstrating food could not act as conditioning stimuli promoting a food preference or avoidance (Supplementary Fig. 6a–c), confirming our previous observations that avoidance towards an unfamiliar odorized food is innately coded (Fig. 2d). Besides, we also observed that this apparent and selective amnesia was independent from food consumption (Supplementary Fig. 7a) or corticosterone elevation. Indeed, mice injected with Cort. i.p. $5.0\,mg\,kg^{-1}$ instead of the associated conditioning stimuli (STFP+/Cort. i.p.) still displayed a food-odorant preference (Fig. 5c). Moreover, we observed a striking resetting of the VS brain area (c-Fos stainings; Fig. 5d, e), that suggests an impairment in the recall memory for a food preference or for its consolidation. Remarkably, as this process was independent from the systemic corticosterone level (Supplementary Fig. 7b, c) and not observed in Axo mice, this cerebral adaptation was thus directly related with the activation of the GG neuronal circuitry itself (Fig. 5d, e).

## Discussion

Acute stress can modulate both the performance of action-outcomes and the learning processes in animals including rodents and humans[24,25]. Impending danger such as the risk of predation increases stress-related hormonal levels in the blood[5,15] and can affect memorization[26]. When faced with imminent danger, alerting senses like the olfactory senses allow the detection of chemical cues signaling for the presence of predators[27]. The olfactory senses are also essential to find food and therefore trade-offs and sorting out of the environmental chemical information has to take place followed by the appropriate behavioral responses. In this study, we investigated the biological relevance of the detection of chemical danger cues by the GG on odor-driven food selection because of its fundamental need for animal survival. We demonstrated that the GG circuitry favors innate odor-driven food decision-making in a context of danger (Fig. 2). Indeed, the preys with a functional GG are able to select the food resource, depending on its smell, while avoiding to be eaten by predators. We have also observed that the GG was not directly implicated in the detection of odorants emitted by food resources (Fig. 2) and in the linked foraging behavior (Fig. 3), we can thus consider that our GG-dependent results are not exclusively specific to food selection[15,17] and that they may be extended to the global effects of threatening scents on odor-driven preferences.

Our understanding of the interplay between the pathways processing GG-induced fear-like responses and those encoding food-odorant information remains to be completed. GG neurons use multi-signaling pathways, principally linked to bitter taste signaling[28] and GCG cascades[18] to detect a large repertoire of chemical danger cues[28,29]. Interestingly, in GC-D neurons, a similar encoding strategy occurs[14] making it also sensitive to fatty acids, steroid hormones as well as to some GG-related cues such as pyrazines. This shared encoding strategy as well as the partial overlap of recognized cues between the GC-D and the GG subsystems might, at the detection level, contribute to the apparent positive collaboration between these two parallel circuitries.

In a pilot experiment performed in the NG complex of GCG–Cre–GFP mice, we noticed the presence of periglomerular cells expressing the tyrosine hydroxylase[30,31] (TH; Supplementary Fig. 8a, b). Interestingly, their density was preferentially located around GC-D glomeruli (GC-D: $9.3 \pm 0.9$ TH + cells, $N_{glomeruli} = 6$; GG: $5.0 \pm 1.2$ TH + cells, $N_{glomeruli} = 7$), supporting the interconnectivity of GC-D glomeruli with other olfactory circuitries such as those of the MOE[32]. On the other hand and compared to other OB regions[33], TH + cell density was moderate around GG glomeruli, which is consistent with their homogenous innervation[19] and their episodic afferent activity[9]. This might question the potential cross-glomerular regulatory task of these periglomerular cells. Moreover, and as a temporal hiatus, we found that the GG circuitry could impact socially learned food preferences in a delayed manner (Figs. 4 and 5), at least 1 h after the STFP procedure (Fig. 4). Therefore, periglomerular regulation emerging from GG on GC-D glomeruli requires further investigations.

The integration of the GG circuit on higher structures in the CNS have revealed specific subregions implicated in fear and anxiety responses such as the posteroventral division of the medial amygdala (MeA) and the dorsomedial subdivision of the ventromedial hypothalamus[16,34] (VMH). Our findings unravel two additional brain domains implicated in this global integration, the APir (Fig. 3) and the VS (Figs. 4 and 5). Interestingly, the APir was previously reported as a specific area of the olfactory cortex that induces stress hormone responses to volatile predator scents[21]. Here, we demonstrate that this heterospecific response could be extended to conspecific-signaling integration, as the mouse alarm pheromone SBT likewise initiates APir activities (Fig. 3). In addition

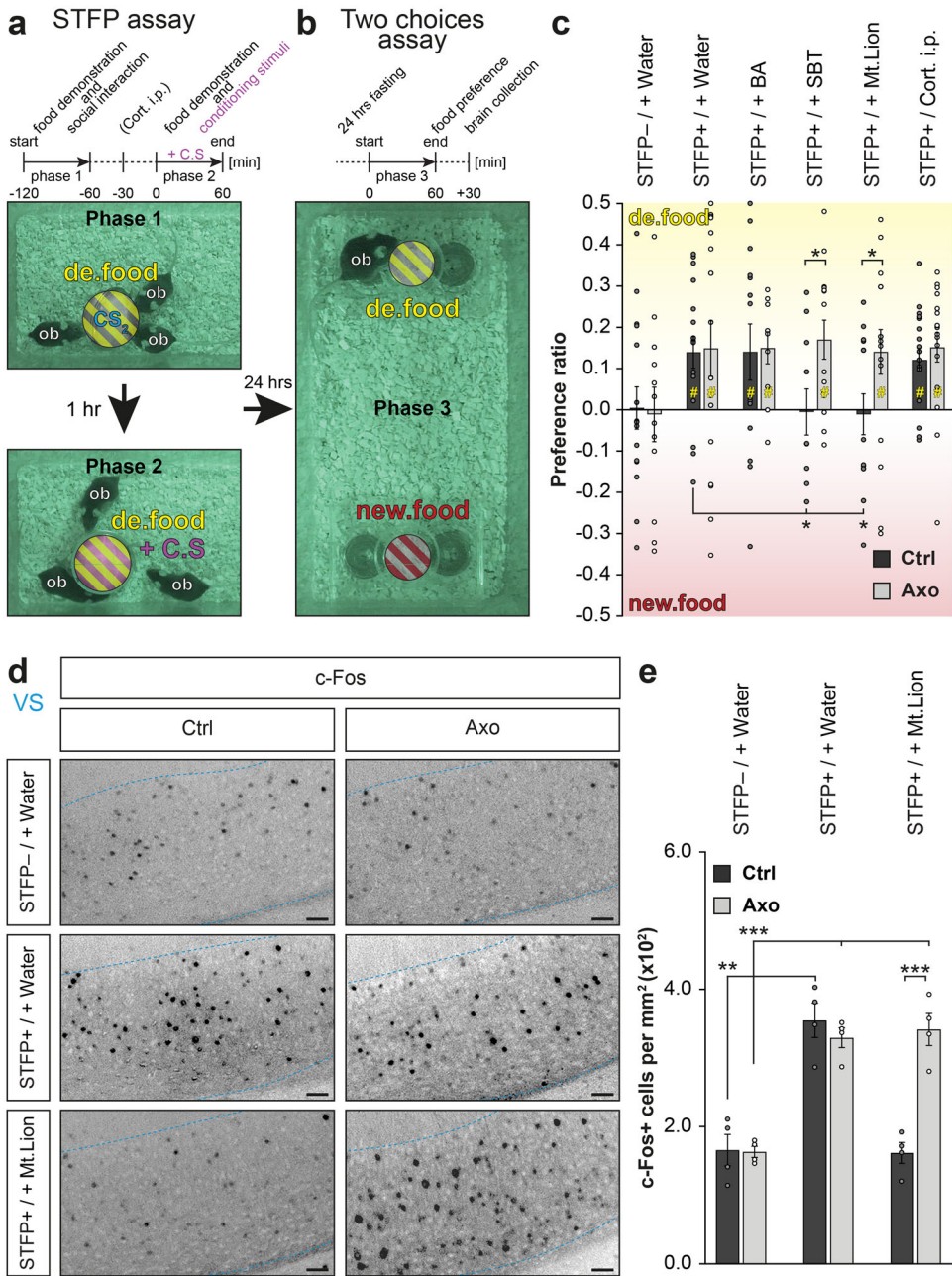

**Fig. 5 Acquisition of a food preference is impaired by environmental GG activation. a** Acquisition of a food preference performed by STFP assay followed by an associated conditioning stimulus (+C.S, purple) is illustrated by infrared snapshots and a procedure time-table. In phase 1, acquisition of a food preference. A demonstrating food (de.food, yellow) is presented to observer mice (ob) with a surrogate display, moistened with CS$_2$, mimicking social interactions. In phase 2, de.food is again presented to ob mice but associated with a +C.S **b** In phase 3, each ob mouse is individually tested in a two choices assay with two sources of food (de.food, yellow; new.food, red). **c** Quantification of the food preference ratio in Ctrl and Axo mice without or with STFP procedure followed by the indicated associated stimulus (STFP− or +/+C.S) or after intraperitoneal injection of 5.0 mg/kg corticosterone (Cort. i.p.). Statistical significances of preference ratio are performed with Z tests, # $p < 0.05$ (yellow, for a preference for de.food); non-significant if not mentioned. 10–18 animals were used per condition. The following pairs of spices were used: cinnamon 1% vs. cocoa 2%; anise 1% vs. oregano 2.4%; thyme 2% vs. basil 1.4%. **d** Representative c-Fos staining (dark spots) in Ctrl and Axo mice under the indicated conditioning in VS (blue dashed line). Scale bars, 50 μm. The following pair of spices was used: nutmeg 1% vs. ginger 1%. **e** Quantification of the density of c-Fos+ cells in VS from (**d**). Four animals were used per condition. **c, e** Values obtained from Ctrl (black) and Axo (gray) mice are represented as mean ± SEM with aligned dot plots. For comparisons between conditions and between Ctrl vs. Axo mice, two-tailed Student's t-tests or Wilcoxon w-tests are used, *$p < 0.05$; **$p < 0.01$; ***$p < 0.001$.

to the instinctive fear response, we, therefore, uncovered here a key consequence of the subsequent systemic corticosterone elevation and demonstrated its central outcome for the improvement of odor-driven food behavior (Fig. 4). The processing ability of the CNS structures, such as the one observed here with the APir is usually dependent of their activation. On the other hand, we found

that the treatment of the information by the VS emerging from the GG circuitry consisted in its ability to reset its memory-retrieval activity for socially learned food response[13,23] (Fig. 5). This last finding is consistent with previous reports implying threatening scents in the impairment of retrieval memory[35] but also supports the VS as a key CNS area implicated in context-dependent stress

and integration of fear memories[36–38]. Moreover, we have observed that the same chemical danger cues can act both during (Fig. 4) and upstream (Fig. 5) of the decision-making event with opposing behavioral outcomes. Threatening scents could thus develop contrasting effects on odor-driven food choices in which the VS should play a context-dependent integrative role. These observations further imply a complex strategy for encoding both the predator-related stress as well as the retrieval of innate and learned fear[39]. At the individual level, this innate trait could be considered as a beneficial mechanism of protection. Indeed, when the food choice is presented in a context of threat (Fig. 4), the favored decision for survival should be to select a safe and familiar food. Nonetheless, to be efficient and to decrease the risk of being eaten, a compromise is made and the emerging strategy is, in the end, to erase this food familiarity and to consume the food anyway (Figs. 3 and 5). Thus, for mice, as for humans, hierarchy of prepotency as well as behavioral motivation is innately coded. Therefore, physiological needs, such as feeding, surpass safety needs[40].

In the wild, the ability to take rapid decisions concerning food choice is critical for the survival of the individual and leads to an increase in the overall fitness of the species. It is therefore conserved throughout the animal kingdom[41,42]. For example, in the nematode *Caenorhabditis elegans* (*C. elegans*), stress linked to environmental conditions can impact food preferences when they are coupled with odorants[43] as well as with alarm pheromones and predator scents[44,45]. Interestingly, this worm modality is dependent on a specific olfactory subsystem composed of a pair of amphid wing cells of type C[46] (AWC). GG neurons were reported to share molecular and functional features with these AWC neurons, conferring to these two olfactory subsystems an interspecies orthologous status[29].

In summary, our present study demonstrates that the GG potentiates, in mice, the decision-making process for the odor-driven food choice under threat. Our results highlight how mice integrate environmental stressful stimuli to control both innate and previously acquired odor-driven food preferences. Olfactory threats are detected by the GG that operates as an immediate sensor acting both on innate perceptions, systemic hormonal levels and on the acquired food preferences to circumstantially revise mice decision-making, a mechanism that confers to the animal a beneficial survival advantage[21].

## Methods

**Animals**. Male and female C57BL/6 (*Mus musculus*; Janvier Labs), OMP-GFP[47] and GCG–Cre–GFP mice[19] (lines 43 and 52) were used. In the gene-targeted mouse strain OMP-GFP, the GFP is used as a histological reporter of mature OSNs[48] expressed under the control of the OMP promoter[47,49]. In transgenic GCG–Cre–GFP mice the expression of GFP is driven by the GCG gene which allows the selective expression of this marker in GG circuitry. Otherwise mentioned, mice were grouped-housed (3–5 animals/cage) at 21.5 °C under a 12 h light/dark cycle with ad libitum access to food (3242; Kliba Nafag) and water. Mice were euthanized by $CO_2$ inhalation and/or by cervical dislocation. The experimental procedures were in accordance with the Swiss legislation and approved by the EXPANIM committee of the Lemanique Animal Facility Network and the veterinary authority of the Canton de Vaud (SCAV).

**Generation of control (Ctrl) and GG Axo mice**. The GG completely degenerated after surgical sectioning of its axonal bundles[50]. Under deep isoflurane anesthesia, GG nerves from mice pups were Axo by using disposable 26 G needles[9]. Similar surgical procedure with only superficial sectioning was used to generate Ctrl mice in which the axotomy was not effective. Axo and Ctrl mice were kept grouped with their littermates until behavioral assays. The efficiency of the GG degeneration was assessed at the end of the behavioral assays using a histological approach. In brief, after fixation in 4% paraformaldehyde (PAF 4%; pH 7.4), mice noses were included in low melting 4–5% agar and serial coronal slices of 80 μm were generated with a vibroslicer (VT1200S, Leica). Slices of GG region were selected under a stereomicroscope (M165 FC, Leica) in accordance with their general morphology[51]. Using a floating immunohistochemical technic[29], presence of GG neurons was detected thanks to their OMP expression. For that, slices were blocked for 3 h in a PBS solution containing 5% normal rabbit serum (NRS; Jackson ImmunoResearch)

and 0.5% triton X-100. The primary antibody (Goat anti-OMP; Wako; 1:1000) was applied to the slices overnight in a PBS solution containing 2.5% NRS and 0.25% Triton X-100. Slices were then washed in 1% NRS and incubated with the secondary antibody (FITC-conjugated, Rabbit anti-Goat; Jackson ImmunoResearch; 1:200) in 1% NRS for 2 h. As control, a similar procedure was systematically performed with the MOE after a decalcification treatment (PBS − EDTA 0.5 M, pH 8.0). For double staining with the enzymatic phosphodiesterase 2A (Rabbit anti-PDE2A; FabGennix; 1:400), serum from donkey (NDS; Jackson ImmunoResearch) and appropriate secondary antibodies (FITC-conjugated, donkey anti-Goat; Jackson ImmunoResearch; 1:200; Cy3-conjugated, donkey anti-Rabbit; Jackson ImmunoResearch; 1:200) were used. For analyses, slices were mounted in antifade mounting medium with DAPI (Vectashield, H-1200; Vector Labs). Observations and acquisitions were made by LED-fluorescence microscopy (EVOS *fl*, AMG) and confocal microscopy (SP5; Leica) with ×4 to ×40 objectives. Maximum projection reconstructions were made with Imaris (Bitplane IMARIS 6.3). Mice with an expression of OMP-positive GG neurons were considered as the Ctrl mice and those with the absence of OMP-positive cells as Axo mice. No differences were observed between untreated and Ctrl mice[9].

**Imaging of the NG complex**. Ctrl and Axo females GCG–Cre–GFP[19] mice (lines 43 and 52) were used to investigate the parallel GG and GC-D circuitries. In brief, using the same floating immunohistochemical approach as previously described, serial coronal slices from GG, MOE and NG from olfactory bulbs of respectively, 80, 200, and 120 μm were generated. Slices were blocked in 10% normal goat serum (NGS; Interchim). Primary antibodies (Chicken anti-GFP; abcam; 1:600. Rabbit anti-PDE2A; FabGennix; 1:400) were applied overnight in 5% NGS. The slices were then washed in 2% NGS and incubated with secondary antibodies (Alexa Fluor 488-conjugated, Goat anti-Chicken; ThermoFisher; 1:200. Cy3-conjugated, Goat anti-Rabbit; Jackson ImmunoResearch; 1:200) in 2% NGS. Staining of periglomerular cells was performed with anti-tyrosine hydroxylase antibody (Mouse anti-TH; Immunostar; 1:2000), a molecular marker of periglomerular and short-axon cells[30]. The immediate-early genes c-Fos (Mouse anti-c-Fos; Santa Cruz Biotechnology; 1:250) was used to evaluate the neuronal activity in the NG complex after 1 h of $CS_2$ (10 ppm) stimulation. Detection of mouse antibodies was done with a specific secondary antibody (Alexa Fluor Plus 647-conjugated, Goat anti-Mouse; ThermoFisher; 1:200). The slices were then mounted and counterstained with DAPI (Vectashield, H-1200; Vector Labs). LED-fluorescence microscopy (EVOS *fl*, AMG) and confocal microscopy (SP5; Leica) with ×4 to ×40 objectives were used and reconstructions were made with Imaris (Bitplane IMARIS 6.3). Quantifications of periglomerular cells or c-Fos-positive cells were performed according to the reconstitution of a section of 15 μm of thickness per glomerulus. A radius of 50 μm around each reconstituted glomerulus was used as a criterion of association with TH-positive or c-Fos-positive cells. Accordingly, a TH-positive or c-Fos-positive cell could be associated with more than one glomerulus.

**Chemicals and spices**. Olfactory stimulations with synthetic cues, predator urine and spices were used during calcium imaging and behavioral assays at the indicated dilutions. BA, mix of pyrazines (2,6-dimethylpyrazine, 2-ethyl-3,5-dimethylpyrazine and 2,3,5-trimethylpyrazine), 2PT, TMT, SBT, carbon disulfide ($CS_2$) were purchased from Sigma-Aldrich, Alfa Aesar and Contech at the highest available purity or synthetized in-house[15,20]. Mt.Lion urine was obtained from PredatorPee and was previously analyzed for its fear-inducing properties[20]. Commercial spices from McCormick or from local distributors were used as infused solution or as dry powders. For calcium imaging and standard two choices assay, liquid stocks of infused spices were previously prepared in double distilled sterile water (50%, weight/vol) during 1 h under constant agitation, filtered (0.22 μm) and stocked at 4 °C. Similar procedure was done for perfusion of powdered food for calcium imaging. For a two choices assay associated with STFP procedures, spices were used as dry powders freshly added to rodent's powdered food (weight/weight)[52–56]. They were coupled as tested paired spices (spice #1 vs. spice #2), in counterbalanced mode to avoid any innate preference[13]; cinnamon (1%) vs. cocoa (2%), anise (1%) vs. oregano (2.4%), thyme (2%) vs. basil (1.4%) and nutmeg (1%) vs. ginger (1%).

**Live calcium imaging**. Thanks to the GG expression of their fluorescent reporter gene, OMP-GFP mice were used for calcium imaging experiments on GG neurons[15]. In brief, mice noses were prepared in ice-cold oxycarbonated artificial cerebrospinal fluid (ACSF; 118 mM NaCl, 25 mM NaHCO₃, 10 mM D-glucose, 2 mM KCl, 2 mM MgCl₂, 1.2 mM NaH₂PO₄, and 2 mM CaCl₂; pH 7.4) under a fluorescence-equipped microscope (M165 FC; Leica). Coronal slices from mice GG of 80 μm were generated on ice with a vibroslicer (VT1200S, Leica) and loaded in a Fura-2 acetoxymethyl ester (AM) (5 μM; TEFLabs) solution containing pluronic acid (0.1%; Pluronic F-127, Invitrogen) during 60 min at 37 °C (5% CO₂). Loaded slices were immobilized in a bath chamber (RC-26, Warner Instruments) with a slice anchor and continuously perfused with ACSF. A bipolar temperature controller (SC-20/CL-100, Warner instruments) was used to maintain the bath temperature between 23 °C and 25 °C. Solutions of perfusion were directly diluted from their stock in ACSF and their final osmolarities were between 285 and 300 Osm/L. Short extracellular perfusions of potassium (KCl; 25 mM) were used as a cellular

viability test. Spices (1:100), $CS_2$ (10 ppm), 2PT (1:5000), SBT (1:5000) and Mt.Lion urine (1:500) were freshly prepared before each experiment. Intracellular calcium variations ≥10% of the endogenous baseline activity were considered as GG-evoked responses[15]. Visualizations and acquisitions with a ×40 objective and a sensitive camera (Cool SNAP-HQ2, Photometrics) were made under an inverted fluorescence microscope (Axio Observer.A1, Zeiss). The software MetaFluor (MetaFluor, Visitron Systems) was used to monitor intracellular calcium variations[57].

**Behavioral assays.** All behavioral assessments were conducted in a dedicated room under the supervision of the affiliated investigator[58]. Adult (8–16 weeks) C57BL/6 mice, with an equivalent sex ratio per conditions, were tested during the nocturnal period of the 12-h light/dark cycle. Tested odorants were only used for a single behavioral trial[59]. Delivery of powdered food was performed with homemade-ballasted cups limiting the food dispersal.

*Two choices assay and quantification of the food preference ratio.* To verify if the mice display a preference between two sources of odorized food (food odor #1 or food odor #2), we adapted previously designed two choices assays[60–62]. In brief, grouped-housed mice were food habituated for 72 consecutive hours with access to 2 g/day/mouse of powdered ordinary food followed by 24 h of food deprivation. After this period of habituation, a food preference test was performed on individual animal. For that, mice could choose, during 1 h, between two cups of odorized powdered food placed in the opposite part of the cage in an accessible semi-closed box (~500 cm$^3$) to limit odorant diffusion. Each source of food was offered in sufficient amount (between 3.0 and 3.5 g). Quantification of the food preference ratio for the food odor #1 was calculated according to the ratio between the food odor #1 consumed versus the total food consumed (food odor #1 + odor #2) minus 0.5, corresponding to the non-preference threshold. Values are then expressed between 0.5 and −0.5 where positive scores correspond to a preference for the food odor #1, negative scores for the food odor #2 and zero correspond to no preference displayed. Values obtained for a consumption ≥0.2 g were processed. To mimic food soiled by different odorants, 500 μl of the investigated substances, diluted in water, were deposited around the food without any direct contacts[60,61]. We used as tested odor #1, distilled water (Water), Cinnamon and Cocoa (10%, 1:5 from liquid stock), $CS_2$ (10 ppm), pyrazines (10%), 2PT (10%), TMT (10% to 0.1%), SBT (10 to 1%) and pure Mt.Lion urine. The odorless Water or the aversive BA[60,63] (10% or adapted) were used as odor #2.

*STFP assay.* Our STFP assay was designed according to previous reports[10,13]. For that, a demonstrator mouse (de) was first separated from its group of observer mice (ob) and fed during 1 h a spiced demonstrating food (de.food). A total consumption of ≥0.2 g by the demonstrator mouse was necessary to proceed. Then, this demonstrator mouse was returned with its littermates for 1 h of social interactions that allowed the acquisition of food preference for this de.food[13]. Depending on our investigations, STFP phases could be condensed by using a surrogate demonstrator mouse presented for 1 h to the observed mice[13]. This surrogate demonstrator mouse was composed of a cotton ball on which the unreachable de.food was supplemented with 500 μl of $CS_2$ (10 ppm) and enclosed in a wire mesh. To verify the acquisition of food preference for the de.food, a two choices assay was then performed after a food diet of 24 h. For that, individual food preference of the observer mice was performed between the de.food (as food odor #1) and an unfamiliar paired spiced food (new.food, as food odor #2). Quantification of the preference ratio for the de.food was then calculated. Demonstrated spices were counterbalanced within pairs in each behavioral trial and cup positions varied randomly. A couple of spices were used only once per mouse and mice could be tested with different sets of spices. Values obtained for a consumption ≥0.2 g were processed. To modulate the STFP performance, the addition of conditioning stimuli (+C.S) were considered during or before the food selection test. Respectively, 500 μl of conditioning stimuli, +Water, +BA (1:500), +SBT (1:500) or +Mt. Lion (pure Mt.Lion urine) were placed around the food during the two choices assay or were coupled for 1 h with the de.food, 1 h after food preference acquisition (STFP assay).

**Systemic corticosterone measurement and injection.** After powdered food habituation and 24 h of fasting, grouped of mice were challenged for 1 h with 500 μl of conditioning stimuli, +Water, +SBT (1:500) or +Mt.Lion (pure Mt.Lion urine). Each stimulus was delivered on a cotton ball containing unreachable powdered food enclosed in a wire mesh. The quantification of the freezing time and the risk assessment behavior displayed by mice[15] was performed for the first 15 min and respectively expressed as a percentage of time to freeze and as a mean score obtained per minute in a repeated measure design. Directly after euthanasia, blood samples were obtained by cardiocentesi, collected in EDTA-coated microtubes (Micro tube 1.3 ml K3E; Sarstedt), centrifuged at room temperature for 5 min at 10,000 × g and plasmas were kept at −80 °C until Elisa analysis[15] (EIA Kit; Enzo Life Sciences). According to the kit manufacturer, corticosterone levels were measured in duplicate and a minimum of four mice were used to evaluate each situation. To mimic the observed increase of stress-related hormonal levels, corticosterone (Corticosterone: HBC complex; Sigma) in 0.9% saline vehicle were

intraperitoneally injected (Cort. i.p., 10.0, 5.0, 2.5, 0.5, 0.0 mg kg$^{-1}$) with 100 μl per 10 g of mice bodyweight 30 min before behavioral assays[64].

**Brain regions and c-Fos analysis.** Thirty minutes after behavioral assays, mice brains were collected and fixed (PAF 4%). Brains were post-fixated overnight in 1% PAF at 4 °C and rinsed in PBS before the inclusion in 4% low melting agar. Coronal slices of 60 μm were generated with a vibroslicer (VT1200S, Leica) on ice-cold PBS and according to their morphology[65], slices comprised between Bregma −3.28 to −3.80 were selected under a stereomicroscope (M165 FC, Leica) and were used for the ventral subiculum (VS) analysis. Slices corresponding to Bregma −3.28 were used for the amygdalopiriform transition area (APir) analysis. A similar immunohistochemical process than the one used before was performed to examine the brain immediate-early genes c-Fos labeling. In brief, selected brain slices were incubated for 60 h with the primary antibody (Rabbit anti-c-Fos; Santa Cruz Biotechnology; 1:1000) in a normal goat serum (NGS; Interchim) solution and the detection was done with a fluorescent secondary antibody (Cy3-conjugated, Goat anti-Rabbit; Jackson ImmunoResearch; 1:200). LED-fluorescence microscopy (EVOS fl, AMG) with ×4 objectives and confocal microscopy (SP5; Leica) with ×20 objective were used for the acquisitions of respectively the VS and the APir regions under a black/white contrast. Using the open access imageJ software (National Institutes of Health, 1.48 v), a stereological counting approach was performed for each investigated brain (4–6 per condition). The mean density of c-Fos positive cells was measured per slice according to 2–4 surfaces per area. Average of a minimum of two slices was used for the establishment of the global density of c-Fos positive cells per area and brain.

**Statistics and reproducibility.** The open-source package R version 3.1.2 was used for statistical analysis and GraphPad Prism 8.2.0 to compute bar graph with their corresponding aligned dot plots. Values are expressed as mean ± standard error of the mean (SEM). Food preferences were assessed by Z-tests according to their calculated mean z values (preference ratio/SEM). Shapiro–Wilk tests and Fisher F-tests were used for evaluation of normality and homoscedasticity. Accordingly, comparisons were performed with the two-tailed Student's t-tests or Wilcoxon (Mann–Whitney) w-tests and no corrections were applied to compare multiple data. Significance levels are indicated as follows for the preference ratio: # $p < 0.05$, in yellow for a significant preference for odor #1 or de.food and in red for a significant preference for odor #2 or new.food; non-significant if not mentioned. For comparisons between Ctrl and Axo mice and between conditions: *$p < 0.05$; **$p < 0.01$; ***$p < 0.001$.

**Reporting summary.** Further information on research design is available in the Nature Research Reporting Summary linked to this article.

## Data availability

All data and materials used in the analysis are available in the main text, in the Supplementary Information and in the Supplementary Data 1 file.

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

## Acknowledgements

We thank I. Rodriguez and F. Resende for the GCG–Cre–GFP mice; S. Kellenberger, R. Stoop and C. Lopes for fruitful discussions on the manuscript. This work was supported by the Department of Pharmacology and Toxicology, University of Lausanne, and by the Swiss National Science Foundation Grant 310030_185161 (to M.-C.B.).

## Author contributions

J.B. designed the project. J.B., A.dV., D.W., and M.N.T carried out experimental procedures. All authors discussed the results and analyzed data. J.B. and M.-C.B. wrote the manuscript.

## Competing interests

The authors declare no competing interests.
