## [Peer Review File · Communications Biology]

Reviewers' comments:

Reviewer #1 (Remarks to the Author):

Brechbuhl et al.

Animals compromise between the environmental risks including predator dangers and the demand for food for their energy needs. Therefore, it is important to optimize their decision-making concerning food selection under threat conditions. In this study, Brechbuhl et al. investigated mice's selection of food scented with many odors including several predator-derived odors, and found that a specific olfactory subsystem, the Gruenberg ganglion (GG), is involved in the selection of food scented with predator odors.

I agree that this manuscript contains interesting findings including the effect of predator odors on the preference of familiar or conspecific odors. The authors also clearly showed that some effects of predator odors are mediated by GG as they previously showed (Brechbuhl et al. 2008 and 2013). However, I have a large concern that the effect of predator odors investigated in this study is not specific to food selection. For example, in Fig.4, the authors showed that STFP is enhanced by predator odors. However, even if they used filter papers in place of foods in this experiment, predator odor may increase the investigation time for "demonstrating filter paper" with an odor presented by demonstrator mice before. If so, the results indicate that predator odors simply affect animal preference to odors presented with their conspecifics. The same can be said of the experiments in Fig. 5. Therefore, I strongly suggest that the authors rewrite the manuscript to describe the global effects of predator odors on odor preferences of mice, or perform additional experiments to show that the effects are food selection-specific.

Since I think that this manuscript contains interesting findings, I would be happy to review the authors' revised manuscript as a new paper.

Reviewer #2 (Remarks to the Author):

Dr. Christine Broillet group (University of Lausanne) has published many important findings regarding ion channels and membrane receptors in olfactory neurons. In particular, her group has recently identified interesting roles and mechanisms of Grueneberg ganglion, one of the olfactory subsystems.

Here, authors tried to suggest another mechanisms which odors closely related to survival (innate odors) could be processed through a circuit connecting the Grueneberg ganglion (GG) area to the necklace glomerular complex in the olfactory bulb (OB). To this end, authors generated a model mouse expressing GFP in neurons expressing guanylyl cyclase G in the GG area. Next they tested the preference for odors of predators or rotten food, either in either the presence or absence of the GG neurons (axotomy of GG neurons). They also confirmed that social interaction is critical for the preference of odors in the group. And the social interaction was strengthened due to increase of corticosterone upon exposure to the odors.

The manuscript is well prepared, but with some rooms for improvements. In particular, it would be greatly appreciated by researchers in the field of olfaction if authors provide enough backup results regarding circuit-behavior and its causality of biological factors. Thus, authors are asked to respond to several issues listed here.

1. Figure 1d, Supplementary 1a : authors can consider to move Figure 1D to the supplementary section, then add theoretical backgrounds regarding the role of short axon cells in the necklace glomeruli in Discussion section. If authors decide to leave Figure 1D in the main figure, authors should show the direct correlations among the short axon cell expressing TH (identified in Figure 1D), the GG-GFP neuron and GC-D neuron expressing PDE2A shown in Figure 2.

2. Figure 1f : authors should provide results of the experimental control of NG + CS2.
3. Figure 2d and 2e : authors should explain the meaning of the preference ratio indicated in the y-axis. What are the criteria for robust adversity and preferences mentioned in the manuscript? Is there any objective preference value in the Ctrl preference ratio for each experimental odor based on the control odor cue (water, butyric acid (BA))? If there is any correlations, what is the max and min scale of the value?
4. Figure 2e : authors described the results of pyrazines, 2PT, TMT, SBT, and Mt.Lion in the axo model as 'reversed' (line 134,). Authors can consider to describe that the preference for BA (in other words, avoidance about threatening odor) disappeared (not flipped). In the case of the reversed pattern used in the text, it should have been shown that the axo model is changed to have a preference for the tested odorant or an avoidance for BA.
5. Figure 3, 4 and 5 : As the result of c-fos expression in amygdalopiriform (Apir) and ventral subiculum (VS) regions dependent of either the presence or absence of corticosterone, authors should show whether the behavioral effects of corticosterone in the axotomy model correlate with the neurons in the region of authors' interest.
6. Discussion : it would be better if authors explain the relationship by adding theoretical backgrounds from previous reports regarding physiological and behavioral correlations between GG olfactory circuit and activation of Apir and VS domains.

Besides, please check for typos correction.
eg. typo in line 146 : alas (?)

Reviewer #3 (Remarks to the Author):

This manuscript elucidates the role of the Grueneberg ganglion (GG) olfactory subsystem by selectively removing it and then assessing odor preferences (novel vs familiar, predator odor vs neutral odor vs other aversive odor), food consumption and corticosterone levels. They convincingly report the selective removal of the GG neurons by axotomy, which does not influence the function of the GC-D subsystem of the MOE (mediating socially acquired food preference via CS2 detection), although target glomeruli are in close proximity. These experiments form the basis of the subsequent comparisons between control and axotomized animals. The differences reported (and absence thereof) are clear and convincing. Predator odors signal via the GG, in contrast to other aversive odors which get processed in the MOE. Predator odors dominate over other aversive odors via GG signalling. Predator odors during food choice enhance the preference for familiar food via increase in systemic corticosterone levels which is caused by GG activation. Socially learned food preference can be extinguished by presentation of predator odor prior to food choice, but only in mice with functioning GG.

This group has done seminal work in the study of the GG subsystem. This manuscript extends these analyses in essential ways and thus advances the field significantly. My main concern is the presentation of the results. The writing is often unclear, and sometimes misleading, a very smooth writing style notwithstanding. The discussion of the somewhat opposing results of figure 4 and 5 could be improved.

Some examples are detailed below.

Line 30-32, Abstract „We also find that the increase in corticosterone, which is GG-dependent, dynamizes safe food preference acquired during social transmission. Moreover, we demonstrate that the memorization of a food preference can be rectified by activation of the GG circuitry..”

I do not think these sentences are understandable. As far as I can tell, the authors use 'dynamize' (only used in Abstract and Introduction, not in Results) to describe a more pronounced preference for safe (socially accepted) food, when GG is simultaneously activated by predator odor. Maybe use 'enhance'?

How can a memory be rectified? Again the term is not used in Results, only in Abstract and Introduction. The reference given for 'memorization' deals with memory retrieval.

Line 89 Cracking the threatening code of food by GG detection.

Sounds impressive but is not very clear. Maybe it would be better to write something like „GG encodes predator odors and dominates over MOE-transmitted odor signals“

Line 250-253

I do not see how the speculation mentioned follows from the results reported here.

Responses to reviewers:

COMMSBIO-20-0343 (The olfactory Grueneberg ganglion drives food choices under threat).

Reviewer #1:

We would like to thank the reviewer for her/his comments. We truly appreciate her/his insightful review of our manuscript. We have now addressed all of her/his questions and suggestions and we believe that taking them into account significantly strengthen our revised manuscript.

In summary, in order to take into account all the comments from the three reviewers, we have now modified the manuscript as follows: We have described the observed effects of threatening scents on *odor-driven food choices* rather than on *food preference*. We have now performed additional control experiments to complete and strengthen our previous observations and accordingly, we have redesigned the Figure 1 and the Figure 3a. We created new Supplementary figures 3 and 8. We also redesigned Supplementary figure 1 and the original Supplementary figures 4 and 6 (our new Supplementary figures 5 and 7). We have now detailed the meaning of the preference ratio in the main text. As suggested by the reviewers we have modified our discussion section to include the important and previously missing notions on the global integration of the GG circuitry (detection, bulbar / CNS integration and output; adding also new references). We have also addressed the contrasting findings of Figures 4 and 5. We have now carefully edited the manuscript for typos and for its clarity and readability.

We hope that these changes will allow a better understanding and interpretation of our results and we thus propose to also modify the title of our manuscript accordingly. We look forward to the reviewer response and we will be glad to respond to any further questions and comments that she/he may have.

Please find here the responses to her/his specific comments:

1) I have a large concern that the effect of predator odors investigated in this study is not specific to food selection. ... I strongly suggest that the authors rewrite the manuscript to describe the global effects of predator odors on odor preferences of mice, or perform additional experiments to show that the effects are food selection-specific.

Answer: We would like to thank the reviewer for this important comment and apologize for the lack of clarity. Previous reports had demonstrated that GG-related ligands (predator scents and rodent alarm pheromones) indeed act on odor preferences (Brechtbühl et al., PNAS, 2013; Debiec et al., PNAS, 2014). We agree with the reviewer that the terminology we used was misleading. We have now indicated in the text that it is the odorants emitted from the food that drive the behavioral motivation state for the animal (*line 162-164*):

“This behavior also highlights that odorants emitted by food were sufficient to initiate this motivational-related behavioral process independently from GG detection (Ctrl, $N_{mouse} = 12$; Axo, $N_{mouse} = 12$; one-tailed w -test: $p = 0.483$, ns)”.

To visually reinforce this notion, we have now also adapted the *panel a* of the *Figure 3* by highlighting the localization of the *unreachable food (food)*.

Original Figure 3a

Revised Figure 3a

Moreover, we paid attention in the text, where the specificity of threatening scents on food selection was indicated and have carefully modified our description to a more global effect on odorants associated to food, for example, by the use of “*odor-driven food choices*”, “*food-odorants*” or “*odorants emitted by food*”. Furthermore, we have now insisted on this relevant point in the discussion section in (*line 242-253*):

“The olfactory senses are also essential to find food and therefore trade-offs and sorting out of the environmental chemical information has to take place followed by the appropriate behavioral responses. In this study, we investigated the biological relevance of the detection of chemical danger cues by the GG on odor-driven food selection because of its fundamental need for animal survival. We demonstrated that the GG circuitry favors innate odor-driven food decision-making in a context of danger (Fig. 2). Indeed, the preys with a functional GG are able to select the food resource, depending on its smell, while avoiding to be eaten by predators. We have also observed that the GG was not directly implicated in the detection of odorants emitted by food resources (Fig. 2) and in the linked foraging behavior (Fig. 3), we can thus consider that our GG-dependent results are not exclusively specific to food selection^{15,17} and that they may be extended to the global effects of threatening scents on odor-driven preferences.”.

Finally, to avoid any confusion, we now propose a new title for our revised manuscript:
“The Grueneberg ganglion controls odor-driven food choices under threat”.

For exhaustive details of our rewriting, please find in the following pages the previously submitted main text annotated, in blue, with our modifications and in strikethrough red for deleted parts:

Original submission # COMMSBIO-20-0343

Title

~~The olfactory Grueneberg ganglion drives food choices under threat~~

The Grueneberg ganglion controls odor-driven food choices under threat

Authors

Julien Brechbühl^{1,2}, Aurélie de Vallière^{1,2,†}, Dean Wood^{1,2,†}, Monique Nenniger Tosato¹ and Marie-Christine Broillet^{1,2,*}.

¹ Department of Pharmacology and Toxicology, Faculty of Biology and Medicine, University of Lausanne, Bugnon 27, CH-1011 Lausanne, Switzerland.

² Department of Biomedical Sciences, Faculty of Biology and Medicine, University of Lausanne, Bugnon 27, CH-1011 Lausanne, Switzerland.

* Correspondence to: Marie-Christine Broillet, Department of Biomedical Sciences, Faculty of Biology and Medicine, University of Lausanne, Bugnon 27, CH-1011 Lausanne, Switzerland. **Email:** mbroille@unil.ch

ORCID to Marie-Christine Broillet: 0000-0002-0487-6638.

ORCID to Julien Brechbühl: 0000-0002-2335-3058.

† These authors contributed equally.

Abstract

The ability to efficiently search for food is fundamental for animal survival. Olfactory messages are used to find food while being aware of the impending risk of predation. How these different olfactory clues are combined to optimize decision-making concerning food selection remains elusive. Here, we find that chemical danger cues drive the food selection in mice via the activation of a specific olfactory subsystem, the Grueneberg ganglion (GG). We show that a functional GG is required to decipher the threatening quality of an unfamiliar food. We also find that the increase in corticosterone, which is GG-dependent, **dynamizes enhances** safe food preference acquired during social transmission. Moreover, we demonstrate that ~~the memorization of a food preference can be rectified~~ **memory retrieval for food preference can be extinguished** by activation of the GG circuitry. Our findings reveal a key function played by the GG in controlling contextual food responses and illustrate how mammalian organisms integrate environmental chemical stress to optimize decision-making.

Introduction

Animals continuously assess the environmental risks and, for survival, may need to compromise between the gain of feeding and the potential cost of exposure to danger¹. Especially, when confronted to food scarcity, behavioral trade-offs need to be made^{2,3}. Olfactory senses play an important role in this process as they help to find food but also to sense and avoid threatening situations⁴⁻⁹ such as predation. For rodents, the attraction to food is primarily determined by familiarity, as scents of known food are innately favored^{2,10}. These scents are detected by olfactory sensory neurons (OSNs) found in the main olfactory epithelium (MOE; Fig. 1a)^{11,12}. Food familiarity can be acquired via communication between individuals, by the so-called “social transmission of food preference” (STFP), a behavioral process that occurs by sniffing the breath of conspecifics and making an association between the novel food scent and the presence of the endogenously-produced carbon disulfide gas (CS₂)^{4,13,14}. For this process to occur, the concomitant recognition of the food odorant by OSNs and the detection of the CS₂ by guanylyl cyclase-D neurons (GC-D; Fig. 1a) of the MOE is essential. On the other hand, odors that signal an impending danger are involuntarily left behind by predators^{15,16} or emitted by stressed congeners as warning alarm pheromones^{9,17}. We showed previously that these danger and warning cues are detected in

a specialized olfactory subsystem, the Grueneberg ganglion^{9,15} (GG; Fig. 1a). They generate, in the recipient mouse, stereotypical fear-related behaviors, such as freezing and risk-assessment as well as stress-related systemic responses^{15,18}.

In this study, we provide evidence and new biological insights on how mice exploit threatening scents to optimize and control their food selection. Using gene-targeted, surgically treated and sham-operated mice as well as a series of integrative behavioral assays, we functionally determine that the GG olfactory subsystem controls innate and acquired food preferences when mice smell an impending danger. The GG is indeed required to decipher the threatening quality of an unfamiliar food. We also find that its activation increases the systemic corticosterone level which in turn **dynamizes** **enhances** safe food preferences acquired during social transmission. Moreover, we finally demonstrate that the acquisition of a food preference itself could be circumstantially **rectified** **reset** by the activation of the GG circuitry, providing a **survival** **decision-making** advantage for mice.

Results

Selective deletion of the GG circuitry. The conflicting environmental olfactory messages carried by **odorants emitted by** familiar food **odorants** and by olfactory danger cues need to be continuously evaluated for risk taking and final food decision-making. To address the functional relevance of the GG in this last process in mice, we first surgically disconnected it by nerve axotomy⁹ (axo; Fig. 1a). Thanks to the expression of the olfactory marker protein (OMP; Fig. 1b and c), a neuronal marker for both the GG and the MOE, we then verified that the axotomy induced specific loss of the GG in axotomized (Axo) but not in control sham-operated mice (Ctrl) (Fig. 1b) without affecting the MOE (Fig. 1c). We next demonstrated that the GC-D neurons were still present in the MOE after GG axotomy (Fig. 1c) thanks to the expression of the enzyme phosphodiesterase 2A (PDE2A; Fig. 1b and c) which is shared by GG and GC-D circuitries¹⁹. We then used transgenic guanylyl cyclase-G (GCG, a marker of GG circuitry)–Cre–green fluorescent protein (GFP) mice¹⁹ to selectively trace the GG connections into their olfactory bulb target (OB; Fig. 1a), the necklace glomerular complex (NG; Fig. 1a). Thereby, we distinctly differentiated, in the NG, the parallel connections between GG and GC-D circuitries and we found **interglomerular cells expressing the tyrosine hydroxylase³⁰ (TH; Fig. 1d and Supplementary Fig. 1a) suggesting potential cross-**

~~interactions. Nevertheless, we next observed, in GCG-Cre-GFP Axo mice,~~ that the GC-D circuitry and, in particular its associated necklace glomeruli, were still intact **in GCG-Cre-GFP Axo mice** (Fig. 4e 1d and Supplementary Fig. 4b 1; **Ctrl, $N_{\text{mouse}} = 3$; Axo, $N_{\text{mouse}} = 5$**). Moreover, this GC-D circuitry was still active in the presence of CS₂, **independently from the GG axotomy procedure,** as verified by immediate early gene c-Fos expression¹⁵ (Fig. 4f 1e and f; ~~Ctrl, $N_{\text{mouse}} = 3$; Axo, $N_{\text{mouse}} = 5$~~ **Ctrl, $N_{\text{glomeruli}} = 6$; Axo, $N_{\text{glomeruli}} = 7$; one-tailed t -test: $p = 0.272$, ns**). We have then systematically used the selective deletion of this olfactory subsystem to functionally dissect the relevance of the GG circuitry in **odor-driven** food selection when mice were exposed to chemical danger cues.

Cracking the threatening code of food by GG detection.

GG encodes threatening scents and dominates over MOE-transmitted odor signals.

Mice naturally disregard unfamiliar food based on the unknown odorants it releases, moreover familiar food soiled with danger cues is likewise avoided². As a first physiological assay, we therefore tested odorants commonly used to odorize food such as non-synthetic spices¹³ and we found that these odorants were not directly detected by GG neurons. Indeed, we revealed with calcium imaging experiments performed on acute Fura-2 acetoxymethyl ester (Fura; Fig. 2a) loaded GG slice preparations from transgenic OMP-GFP mice⁹ that, in a total of 54 living GFP tagged GG neurons ($N_{\text{mouse}} = 4$; $n_{\text{slice}} = 6$), Cinnamon, Cocoa, Anise, Oregano, Thyme, Basil, Nutmeg, Ginger as well as the standard mice food and CS₂ did not generate any GG neuronal activity (Fig. 2b). On the other hand, the predator-derived cues 2-propylthietane (2PT) from the stoat anal glands, the mouse alarm pheromone 2-sec-butyl-4,5-dihydrothiazole (SBT) as well as mountain lion (Mt.Lion) urine^{15,20} initiated **strong** reversible calcium responses in respectively 83 %, 57 % and 100 % of the GG neurons. Thus, this first approach not only suggest that the unfamiliarity of a food encoded by its emitted odorants could not be directly deciphered by the GG but also confirmed the ability of the GG to identify danger cues potentially emitted by soiled food.

We next verified, in an integrative context of food choice, the implication of the GG in decoding food unfamiliarity **based on the odorants emitted**. Ctrl and Axo mice were challenged to select, between familiar and unfamiliar food in a two choices assay (Fig. 2c and Supplementary Fig. 2a). To that purpose,

two familiar powdered foods were proposed to mice, odorized either with a series of never-encountered before odorants (odor #1; as unfamiliar food) or with the odorless water (odor #2; as familiar food). We placed all the tested odorants around the powdered foods to exclude any toxicity potentially displayed by synthetic cues. Subsequently, the preference for an odorized food was calculated as the ratio between the consumption of food odor #1 versus the total food consumed (food odor #1 + odor #2) in which the 0.5 value, corresponding to the non-preference threshold was subtracted. Values were thus expressed between 0.5 to -0.5 where positive scores corresponded to a preference for the food odor #1; negative scores for the food odor #2 and zero corresponded to no preference displayed. Hence we showed that rodents indeed prefer familiar food² as non-synthetic spices such as the unfamiliar Cinnamon or Cocoa¹³ as odor #1 were not preferred by mice. Moreover, we observed that this innate choice was also performed by Axo mice (Fig. 2d), confirming that this avoidance behavior was indeed not directly dependent of the GG detection (Fig. 2a-b) but also pointed out to the conserved MOE functionality in the GG axotomized mouse model. Then we found that the GC-D-related ligand CS₂, without any social context¹⁰, did not influence diet selection in both phenotypes (Fig. 2d). We next tested butyric acid (BA) a known aversive odorant that smells rancid and has no alerting relevance¹⁵ and found that it indeed generated **strong** food avoidance both in Ctrl and Axo mice confirming its previously reported GG-independent detection¹⁵ (Fig. 2d). Food odorized with predator scents were also **strongly** aversive (Fig. 2d) as observed with the pyrazine analogues (Pyrazines), found for example in Mt.Lion urine²⁰, 2-PT, 2,4,5-trimethylthiazoline (TMT) from the red fox feces as well as the mouse alarm pheromone SBT. Nevertheless, in Axo mice, this dislike was significantly reduced indicating and confirming (Fig. 2b) that the GG was indeed implicated in the perception of these chemical danger cues¹⁵. Finally, we used Mt.Lion urine as a natural source of predator scents²⁰ and found that its aversive effect on food choice was exclusively dependent on a functional GG (Fig. 2d). Thus, confirming our GG calcium imaging investigations (Fig. 2a-b) that show that mice do not recognize **aversive-caution** odor signals emitted by unfamiliar food via GG detection as demonstrated by exposing them to familiar versus unfamiliar foods (Fig. 2d).

We next highlighted that this olfactory subsystem was fundamental for mice to decipher the threatening quality of an unfamiliar food (Fig. 2e). Indeed, on sets of naïve mice, when we used the pungent and unfamiliar BA as odor #2 (Fig. 2e) in the previous two choices assay (Fig. 2c), mice from

both phenotypes now preferred Cinnamon, Cocoa and CS₂ as sources of unfamiliar odorized food (Fig. 2e) ~~demonstrating the robust~~ confirming the previously observed aversivity of the BA (Fig. 2c). Interestingly, ~~this behavior was entirely reversed~~ in the presence of danger cues such as the Pyrazines, 2PT, TMT, SBT and Mt.Lion urine, ~~which were strongly avoided by~~ Ctrl mice ~~that~~ now systematically preferred the aversive and unfamiliar BA (Fig. 2e). TMT and SBT were particularly efficient as they were still able to generate this innate reaction in serial dilutions (Supplementary Fig. 2b and c). Remarkably, this observed ~~innate food selection was impaired~~ preference for BA disappeared in the absence of a functional GG, without affecting the total food consumption (Supplementary Fig. 2d and e). Thus, mice decode the threatening quality of unfamiliar food by GG detection.

Taken together, our results show that, the GG acts as an immediate ~~and essential~~ sensor, which deciphers the threatening quality of odorants emitted by food. Mice will take their final consumption decision about the safety of the resource thanks to its smell and to its GG perception.

Threatening scents activate GG-dependent corticosterone responses. In the wild, food resources are often limited and could ~~alas~~ be located in a dangerous setting such as impending predation. Nevertheless, motivational state is modified under hunger context³. To evaluate trade-offs displayed by fasting mice when confronted to actively searching a food resource in a danger context, we next challenged Ctrl and Axo mice with unreachable food resources moistened with SBT and with Mt.Lion urine, as sources of respectively intra- and inter-species conditioning stimuli (+ C.S) mimicking environmental evidences of olfactory threats (Fig. 3a). Remarkably, we observed that mice displayed risk assessment behavior (Fig. 3a) in both phenotypes¹⁵, indicating that, in a context of food scarceness, the opportunity to eat indeed overrules fear (+ Water: 8.2 ± 1.0 per min; + SBT: 7.2 ± 1.3 per min; + Mt.Lion: 7.6 ± 1.0 per min. $N_{\text{mouse}} = 8$. RM one-way ANOVA: $F_{(2,14)} = 0.9016$, $p = 0.428$, ns). This behavior ~~also highlights that odorants emitted by food were sufficient to initiate this motivational-related behavioral process independently from GG detection (Ctrl, $N_{\text{mouse}} = 12$; Axo, $N_{\text{mouse}} = 12$; one-tailed w -test: $p = 0.483$, ns).~~ We then followed the systemic integration of these stressful situations with the expression of the immediate early gene c-Fos²¹. We focused on the amygdalopiriform transition area (APir; Fig. 3b) which is the brain region implicated in the increase of stress-related hormone level in the blood when mice smell volatile predator scents^{15,22}. We

found that this brain region was significantly activated by both SBT and the predator urine (Fig. 3c-d). We next confirmed this result by measuring an increase of the systemic corticosterone level (Fig. 3e), implying that intra- and inter-species danger cues were both processed in this specific APir nucleus. In a dilution series of intraperitoneal corticosterone injections (Cort. i.p.; Fig. 3f), we were further able to mimic the observed stress-related hormonal elevation with an amount of Cort. i.p. of 5.0 mg kg⁻¹, **therefore bypassing APir activation** in absence of conditioning stimuli (– C.S; Fig. 3f **and Supplementary Fig. 3**). Moreover, we found that, in Axo mice, the activation of the APir region as well as the elevation of the systemic corticosterone were significantly impaired in this threatening context (Fig. 3c to e), demonstrating that, although the GG was not involved **in the risk-taking of mice** in **odor-driven** foraging, it was essential for the hormonal and physiological adaptation to danger sensing.

Elevation of corticosterone optimizes learned-food selection. In the search of food resources, knowledge of familiar food obtained by STFP support the food decision-making by a recall memory process that requires the activation of a specific part of the hippocampus, the ventral subiculum^{13,23} (VS; Fig. 3b). In this study, we found that the presence of danger cues enhanced this learned food-**odorant** preference when we performed a two choices assay under threatening conditions. For that, **we first trained mice to develop a food-odorant preference. Briefly**, a demonstrator mouse **first** ate an odorized demonstrating food (de.food; standard powdered food odorized for example with Cinnamon as a spice #1; Phase 1; Fig. 4a). Then it returned with its observer littermates to allow the STFP learning process to happen (Phase 2; Fig. 4a). Finally, after 24 hours of fasting, the observer mice were individually confronted to a two choices assay between two odorized foods, the demonstrating food and a novel food (new.food; standard powdered food odorized for example with Cocoa as spice #2; Phase 3; Fig. 4b), both surrounded with the same conditioning stimulus (+ C.S; Fig. 4b). To avoid any individual innate preference, spices were used in a counterbalanced mode¹³ (Supplementary Fig. **4a** to c). We observed in both phenotypes that STFP was, indeed, required to develop a food preference (STFP+ | + Water; Fig. 4c) that was, remarkably, not impacted by a chemical confusion generated by the pungent BA (STFP+ | + BA; Fig. 4c). These first effects not only confirmed the conserved functionality of the GC-D circuitry in Axo mice (Fig. 1c **and e-f**) but also pointed out that, **under aversive distraction, mice still olfactively decipher**

food preferences. Surprisingly, we obtained, with the danger cue SBT (STFP+ | + SBT) and with Mt.Lion urine (STFP+ | + Mt.Lion) a significant enhancement of the food-odorant preference (Fig. 4c). Indeed, the observed mean food preferences were amplified by around 60 % compared to STFP under standard environmental (STFP+ | + Water) or under pungent conditions (STFP+ | + BA). This enhancement was observed in Ctrl but not in Axo mice. It was found to be both independent from food consumption (Supplementary Fig. 4 5a) and from an improvement of the recall memory process as no significant increase in c-Fos activity was observed in the VS brain area under these conditions (Fig. 4d and e). Interestingly, we found that injection of Cort. i.p. 5.0 mg kg⁻¹ under neutral context (STFP+ | Cort. i.p.) was sufficient to mimic this behavioral improvement both in Ctrl and Axo mice (Fig. 4c) **without affecting the memory retrieval activity of the VS (Supplementary Fig. 5b and c).** We thus demonstrated here that, when danger cues are sensed by the GG, the increase in the systemic corticosterone level that occurs (Fig. 3e) leads to an enhancement of the food preference previously acquired by STFP.

GG circuitry activation selectively erase safety food memory. Conspecific interactions are useful for mice to get food familiarity. We found here that mice can also benefit from threatening chemical information present in their environment to reset this previously acquired food-odorant familiarity when associated with a danger context. We indeed tested the impact of olfactory threats on the acquisition of a **new food-odorant preference and thus as we** challenged mice 1 hour after an STFP procedure (Phase 1; Fig. 5a) with a surrogate display that contained the unreachable demonstrating food moistened with an associated conditioning stimulus (Phase 2; + C.S; Fig. 5a). After 24 hours of fasting, the observer mice were then tested in a two choices assay (Phase 3; Fig. 5b). Unexpectedly, we found that, in association with the danger cues SBT (STFP+ / + SBT) or with Mt.Lion urine (STFP+ / + Mt.Lion), Ctrl mice did not display food-odorant preferences while they were still observed in Axo mice (Fig. 5c). As a control, we verified that GG-related ligands directly associated with a demonstrating food could not act as conditioning stimuli promoting a food preference or avoidance (Supplementary Fig. 5 6a to c), confirming our previous observations that avoidance towards an unfamiliar odorized food is innately coded (Fig. 2d). Besides, we also observed that this apparent and selective amnesia was independent from food consumption (Supplementary Fig. 6 7a) or corticosterone elevation. Indeed, mice injected with Cort. i.p. 5.0 mg kg⁻¹

instead of the associated conditioning stimuli (STFP+ / Cort. i.p.) still displayed a food-odorant preference (Fig. 5c). Moreover, we observed a striking resetting of the VS brain area (c-Fos stainings; Fig. 5d and e), that strongly suggests an impairment in the recall memory for a food preference or for its consolidation. Remarkably, as this process was independent from the systemic corticosterone level (Supplementary Fig. 7b and c) and not observed in Axo mice, this cerebral reorientation adaptation was thus directly related with the activation of the GG neuronal circuitry itself (Fig. 5d and e).

Discussion

Acute stress can modulate both the performance of action-outcomes and the learning processes in animals including rodents and humans^{24,25}. Impending danger such as the risk of predation increases stress-related hormonal levels in the blood^{5,15} and can affect memorization²⁶. When faced with imminent danger, alerting senses like the olfactory senses allow the detection of chemical cues signaling for the presence of predators²⁷. The olfactory senses are also essential to find food and therefore trade-offs and sorting out of the environmental chemical information has to take place followed by the appropriate behavioral responses. In this study, we investigated the biological relevance of the detection of chemical danger cues by the GG on odor-driven food selection because of its fundamental need for animal survival. We demonstrated that the GG circuitry favors innate odor-driven food decision-making in a context of danger (Fig. 2). Indeed, the preys with a functional GG are able to select the food resource, depending on its smell, while avoiding to be eaten by predators. We have also observed that the GG was not directly implicated in the detection of odorants emitted by food resources (Fig. 2) and in the linked foraging behavior (Fig. 3), we can thus consider that our GG-dependent results are not exclusively specific to food selection^{15,17} and that they may be extended to the global effects of threatening scents on odor-driven preferences.

Our understanding of the interplay between the pathways processing GG-induced fear-like responses and those encoding food-odorant information remains to be completed. GG neurons use multi-signaling pathways, principally linked to bitter taste signaling²⁸ and guanylyl cyclase-G cascades¹⁸ to detect a large repertoire of chemical danger cues^{28,29}. Interestingly, in GC-D neurons, a similar encoding strategy occurs¹⁴ making it also sensitive to fatty acids, steroid hormones as well as to some GG-related

cues such as pyrazines. This shared encoding strategy as well as the partial overlap of recognized cues between the GC-D and the GG subsystems might, at the detection level, contribute to the apparent positive collaboration between these two parallel circuitries.

In a pilot experiment performed in the NG complex of GCG–Cre–GFP mice, we noticed the presence of periglomerular cells expressing the tyrosine hydroxylase^{30,31} (TH; Supplementary Fig. 8a and b). Interestingly, their density was preferentially located around GC-D glomeruli (GC-D: 9.3 ± 0.9 TH+ cells, $N_{glomeruli} = 6$; GG: 5.0 ± 1.2 TH+ cells, $N_{glomeruli} = 7$), supporting the interconnectivity of GC-D glomeruli with other olfactory circuitries such as those of the MOE³². On the other hand and compared to other OB regions³³, TH+ cell density was moderate around GG glomeruli, which is consistent with their homogenous innervation¹⁹ and their episodic afferent activity⁹. This might question the potential cross-glomerular regulatory task of these periglomerular cells. Moreover, and as a temporal hiatus, we found that the GG circuitry could impact socially learned food preferences in a delayed manner (Fig. 4 and Fig.5), at least 1 hour after the STFP procedure (Fig. 4). Therefore periglomerular regulation emerging from GG on GC-D glomeruli requires further investigations.

The integration of the GG circuitry on higher structures in the CNS have revealed specific subregions implicated in fear and anxiety responses such as the posteroventral division of the medial amygdala (MeA) and the dorsomedial subdivision of the ventromedial hypothalamus^{16,34} (VMH). Our findings unravel two additional brain domains implicated in this global integration, the APir (Fig. 3) and the VS (Fig. 4 and Fig. 5). Interestingly, the APir was previously reported as a specific area of the olfactory cortex that induces stress hormone responses to volatile predator scents²¹. Here, we demonstrate that this heterospecific response could be extended to conspecific-signaling integration, as the mouse alarm pheromone SBT likewise initiates APir activities (Fig. 3). In addition to the instinctive fear response, we therefore uncovered here a key consequence of the subsequent systemic corticosterone elevation and demonstrated its central outcome for the improvement of odor-driven food behavior (Fig. 4). The processing ability of the CNS structures, such as the one observed here with the APir is usually dependent of their activation. On the other hand, we found that the treatment of the information by the VS emerging from the GG circuitry consisted in its ability to reset its memory-retrieval activity for socially-learned food response^{13,23} (Fig. 5). This last finding is consistent with previous reports implying

threatening scents in the impairment of retrieval memory³⁵ but also supports the VS as a key CNS area implicated in context-dependent stress and integration of fear memories³⁶⁻³⁸. Moreover, we have observed that the same chemical danger cues can act both during (Fig. 4) and upstream (Fig. 5) of the decision-making event with opposing behavioral outcomes. Threatening scents could thus develop contrasting effects on odor-driven food choices in which the VS should play a context-dependent integrative role. These observations further imply a complex strategy for encoding both the predator-related stress as well as the retrieval of innate and learned fear³⁹. At the individual level, this innate trait could be considered as a beneficial mechanism of protection. Indeed, when the food choice is presented in a context of threat (Fig. 4), the favored decision for survival should be to select a safe and familiar food. Nonetheless, to be efficient and to decrease the risk of being eaten, a compromise is made and the emerging strategy is, in the end, to erase this food familiarity and to consume the food anyway (Fig. 3 and Fig. 5). Thus, for mice, as for humans, hierarchy of prepotency as well as behavioral motivation is innately coded. Therefore, physiological needs, such as feeding, surpass safety needs⁴⁰.

In the wild, the ability to take rapid decisions concerning food choice is critical for the survival of the individual and leads to an increase in the overall fitness of the species. It is therefore conserved throughout the animal kingdom^{41,42}. For example, in the nematode *Caenorhabditis elegans* (*C. elegans*), stress linked to environmental conditions can impact food preferences when they are coupled with odorants⁴³ as well as with alarm pheromones and predator scents^{44,45}. Interestingly, this worm modality is dependent on a specific olfactory subsystem composed of a pair of amphid wing cells of type C⁴⁶ (AWC). GG neurons were reported to share molecular and functional features with these AWC neurons, conferring to these two olfactory subsystems an inter-species orthologous status²⁹.

In summary, our present study demonstrates that the GG potentiates, in mice, the decision-making process for the odor-driven food choice under threat. Our results highlight how mice integrate environmental stressful stimuli to control both innate and previously acquired odor-driven food preferences. Olfactory threats are detected by the GG that operates as an immediate sensor acting both on innate perceptions, systemic hormonal levels and on the acquired food preferences to circumstantially revise mice decision-making, a mechanism that confers to the animal a beneficial survival advantage²¹.

Responses to reviewers:

COMMSBIO-20-0343 (The olfactory Grueneberg ganglion drives food choices under threat).

Reviewer #2:

We would like to thank the reviewer for her/his comments. We truly appreciate her/his insightful review of our manuscript. We have now addressed all of her/his questions and suggestions and we believe that taking them into account significantly strengthen our revised manuscript.

In summary, in order to take into account all the comments from the three reviewers, we have now modified the manuscript as follows: We have described the observed effects of threatening scents on *odor-driven food choices* rather than on *food preference*. We have now performed additional control experiments to complete and strengthen our previous observations and accordingly, we have redesigned the Figure 1 and the Figure 3a. We created new Supplementary figures 3 and 8. We also redesigned Supplementary figure 1 and the original Supplementary figures 4 and 6 (our new Supplementary figures 5 and 7). We have now detailed the meaning of the preference ratio in the main text. As suggested by the reviewers we have modified our discussion section to include the important and previously missing notions on the global integration of the GG circuitry (detection, bulbar / CNS integration and output; adding also new references). We have also addressed the contrasting findings of Figures 4 and 5. We have now carefully edited the manuscript for typos and for its clarity and readability.

We hope that these changes will allow a better understanding and interpretation of our results and we thus propose to also modify the title of our manuscript accordingly. We look forward to the reviewer response and we will be glad to respond to any further questions and comments that she/he may have.

Please find here the responses to her/his specific comments:

1) Figure 1d, Supplementary 1a: authors can consider to move Figure 1D to the supplementary section, then add theoretical backgrounds regarding the role of short axon cells in the necklace glomeruli in Discussion section. If authors decide to leave Figure 1D in the main figure, authors should show the direct correlations among the short axon cell expressing TH (identified in Figure 1D), the GG-GFP neuron and GC-D neuron expressing PDE2A shown in Figure 2.

Answer: We thank the reviewer for this relevant comment. We agree with her/him that no direct correlation between GG and GC-D glomeruli via the identified short axon cells was shown. Consequently, and as suggested by the reviewer, we have now moved the original Figures 1d and regrouped it with the Supplementary figure 1a into a new *Supplementary Figure 8*.

New Supplementary Figure 8

We have now discussed about these periglomerular cells *in the discussion section* and added *theoretical background* concerning their putative role in the necklace complex (*line 263-274*):

“In a pilot experiment performed in the NG complex of GCG–Cre–GFP mice, we noticed the presence of periglomerular cells expressing the tyrosine hydroxylase^{30,31} (TH; Supplementary Fig. 8a and b). Interestingly, their density was preferentially located around GC-D glomeruli (GC-D: 9.3 ± 0.9 TH+ cells, $N_{glomeruli} = 6$; GG: 5.0 ± 1.2 TH+ cells, $N_{glomeruli} = 7$), supporting the interconnectivity of GC-D glomeruli with other olfactory circuitries such as those of the MOE³².

On the other hand and compared to other OB regions³³, TH+ cell density was moderate around GG glomeruli, which is consistent with their homogenous innervation¹⁹ and their episodic afferent activity⁹. This might question the potential cross-glomerular regulatory task of these periglomerular cells. Moreover, and as a temporal hiatus, we found that the GG circuitry could impact socially learned food preferences in a delayed manner (Fig. 4 and Fig.5), at least 1 hour after the STFP procedure (Fig. 4). Therefore, periglomerular regulation emerging from GG on GC-D glomeruli requires further investigations.”

2) Figure 1f: authors should provide results of the experimental control of NG + CS₂.

Answer: We thank the reviewer for this comment. We have now performed additional experiments on control mice under NG + CS₂ and NG + Water conditions. We have included the new data in a modified Figure 1 with the corresponding statistical analysis in panel 1f. We believe that these important control results strengthen our revised manuscript.

3) Figure 2d and 2e: authors should explain the meaning of the preference ratio indicated in the y-axis. What are the criteria for robust adversity and preferences mentioned in the manuscript? Is there any objective preference value in the Ctrl preference ratio for each experimental odor based on the control odor cue (water, butyric acid (BA))? If there is any correlations, what is the max and min scale of the value?

Answer: We agree with the reviewer that this methodological description was lacking from our original main text. Its incorporation is a plus for the general clarity and readability of our manuscript. Accordingly, we have now explained the meaning of the preference ratio in the main text and in the legend of Figure 2 as well as we have indicated the max and min scale of the value (line 110-115):

“Subsequently, the preference for an odorized food was calculated as the ratio between the consumption of food odor #1 versus the total food consumed (food odor #1 + odor #2) in which the 0.5 value, corresponding to the non-preference threshold was subtracted. Values were thus expressed between 0.5 to -0.5 where positive scores corresponded to a preference for the food odor #1; negative scores for the food odor #2 and zero corresponded to no preference displayed.”.

In legend of Figure 2, (line 705-708): *“Quantification of food Preference ratio for Ctrl (black) and Axo (grey) mice, calculated as the ratio of food consumption [(odor #1) / (odor #1 + odor #2)] - 0.5. Positive scores display a preference for the food odor #1; negative scores for the food odor #2.”.*

As mentioned by the reviewer, there is no objective criteria or reference values for Ctrl preference ratio. Thus, we have now systematically removed all adjectives such as “*robust*”, “*strong*” or other descriptive interpretations. **For exhaustive details of our rewriting, please find at the end of our Responses document, the original text annotated, in blue, with our modifications and in strikethrough red for deleted parts.**

4) Figure 2e : authors described the results of pyrazines, 2PT, TMT, SBT, and Mt.Lion in the axo model as 'reversed' (line 134.). Authors can consider to describe that the preference for BA (in other words, avoidance about threatening odor) disappeared (not flipped). In the case of the reversed pattern used in the text, it should have been shown that the axo model is changed to have a preference for the tested odorant or an avoidance for BA.

Answer: We thank the reviewer for sharing her/his concern. We have now modified our description accordingly (line 140-145):

“Interestingly, in the presence of danger cues such as the Pyrazines, 2PT, TMT, SBT and Mt.Lion urine, Ctrl mice now systematically preferred the aversive and unfamiliar BA (Fig. 2e).

TMT and SBT were particularly efficient as they were still able to generate this innate reaction in serial dilutions (Supplementary Fig. 2b and c). Remarkably, this observed preference for BA disappeared in the absence of a functional GG, without affecting the total food consumption (Supplementary Fig. 2d and e).”.

5) Figure 3, 4 and 5 : As the result of c-fos expression in amygdalopiriform (APir) and ventral subiculum (VS) regions dependent of either the presence or absence of corticosterone, authors should show whether the behavioral effects of corticosterone in the axotomy model correlate with the neurons in the region of authors’ interest.

Answer: We would like to thank the reviewer for this important comment. We agree with the reviewer on the importance of these control results and we have now performed these additional experiments.

In our new Supplementary figure 3, we now show that an intraperitoneal injection of 5.0 mg kg⁻¹ of corticosterone indeed bypasses APir activation in absence of conditioning stimuli in both Ctrl and Axo mice. This lack of increase in c-Fos staining in both genotypes thus correlates with the expected activity of the neurons in the APir region and confirm the results we had obtained previously. We have now modified the main text accordingly (line 172-174):

“we were further able to mimic the observed stress-related hormonal elevation with an amount of Cort. i.p. of 5.0 mg kg⁻¹, therefore bypassing APir activation in absence of conditioning stimuli (– C.S; Fig. 3f and Supplementary Fig. 3)”.

New Supplementary Figure 3

In our revised Supplementary figure 5, we now show that an intraperitoneal injection of 5.0 mg kg⁻¹ of corticosterone indeed does not affect the expected activity of the neurons found in the VS region when mice are under STFP condition. Thus, for this paradigm, the observed behavioral effects of corticosterone correlate with the neuronal activity in the VS region that we had found previously (systemic corticosterone does not affect the memory retrieval but the behavioral outcome). We have now modified the main text accordingly (*line 206-208*):

“Interestingly, we found that injection of Cort. i.p. 5.0 mg kg⁻¹ under neutral context (STFP+ | Cort. i.p.) was sufficient to mimic this behavioral improvement both in Ctrl and Axo mice (Fig. 4c) without affecting the memory retrieval activity of the VS (Supplementary Fig. 5b and c).”

Original Supplementary Figure 4

Revised Supplementary Figure 5

In our revised Supplementary figure 7, we now show that an intraperitoneal injection of 5.0 mg kg⁻¹ of corticosterone during STFP assay does not affect the activity of the neurons found in the VS region after a two-choices assay. Thus, the observed behavioral effects of corticosterone correlate with the expected neuronal activity in the VS region. Mice indeed displayed a normal memory retrieval activity. These observations thus confirm our original results for this paradigm (the observed reset of the memory retrieval is not linked to an elevation of the systemic corticosterone). We have now modified the main text accordingly (*line 232-235*):

“Remarkably, as this process was independent from the systemic corticosterone level (Supplementary Fig. 7b and c) and not observed in Axo mice, this cerebral adaptation was thus directly related with the activation of the GG neuronal circuitry itself (Fig. 5d and e).”

Original Supplementary Figure 6

Revised Supplementary Figure 7

6) Discussion : it would be better if authors explain the relationship by adding theoretical backgrounds from previous reports regarding physiological and behavioral correlations between GG olfactory circuit and activation of *Apir* and *VS* domains.

Answer: We would like to thank the reviewer for this important comment and we have now added theoretical background on the global integration of the GG circuitry and in particular the information concerning the *APir* and *VS* domains. To summarize, very limited regions in the olfactory bulb or in the CNS were previously reported to belong to the GG circuitry (*NG*, *MeA* and *VMH*). Moreover, and to our knowledge, our experiments report for the first time that the *APir* and the *VS* region are integrated in the GG circuitry. Accordingly, and as suggested by the reviewer, we have now significantly changed the overall structure of the discussion section by adding both physiological and behavioral relationships between GG circuitry and the activation of *Apir* / *VS* domains at the detection, the bulbar and the CNS level (*line 254-303*):

“Our understanding of the interplay between the pathways processing GG-induced fear-like responses and those encoding food-odorant information remains to be completed. GG neurons use multi-signaling pathways, principally linked to bitter taste signaling²⁸ and guanylyl cyclase-G cascades¹⁸ to detect a large repertoire of chemical danger cues^{28,29}. Interestingly, in GC-D neurons, a similar encoding strategy occurs¹⁴ making it also sensitive to fatty acids, steroid hormones as well as to some GG-related cues such as pyrazines. This shared encoding strategy as well as the partial overlap of recognized cues between the GC-D and the GG subsystems might, at the detection level, contribute to the apparent positive collaboration between these two parallel circuitries.

...

The integration of the GG circuitry on higher structures in the CNS have revealed specific subregions implicated in fear and anxiety responses such as the posteroventral division of the medial amygdala (MeA) and the dorsomedial subdivision of the ventromedial hypothalamus^{16,34} (VMH). Our findings unravel two additional brain domains implicated in this global integration, the APir (Fig. 3) and the VS (Fig. 4 and Fig. 5). Interestingly, the APir was previously reported as a specific area of the olfactory cortex that induces stress hormone responses to volatile predator scents²¹. Here, we demonstrate that this heterospecific response could be extended to conspecific-signaling integration, as the mouse alarm pheromone SBT likewise initiates APir activities (Fig. 3). In addition to the instinctive fear response, we therefore uncovered here a key consequence of the subsequent systemic corticosterone elevation and demonstrated its central outcome for the improvement of odor-driven food behavior (Fig. 4). The processing ability of the CNS structures, such as the one observed here with the APir is usually dependent of their activation. On the other hand, we found that the treatment of the information by the VS emerging from the GG circuitry consisted in its ability to reset its memory-retrieval activity for socially-learned food response^{13,23} (Fig. 5). This last finding is consistent with previous reports implying threatening scents in the impairment of retrieval memory³⁵ but also supports the VS as a key CNS area implicated in context-dependent stress and integration of fear memories³⁶⁻³⁸. Moreover, we have observed that the same chemical danger cues can act both during (Fig. 4) and upstream (Fig. 5) of the decision-making event with opposing behavioral outcomes. Threatening scents could thus develop contrasting effects on odor-driven food choices in which the VS should play a context-dependent integrative role. These observations further imply a complex strategy for encoding both the predator-related stress as well as the retrieval of innate and learned fear³⁹. At the individual level, this innate trait could be considered as a beneficial mechanism of protection. Indeed, when the food choice is presented in a context of threat (Fig. 4), the favored decision for survival should be to select a safe and familiar food. Nonetheless, to be efficient and to decrease the risk of being eaten, a compromise is made and the emerging strategy is, in the end, to erase this food familiarity and to consume the food anyway (Fig. 3 and Fig. 5). Thus, for mice, as for humans, hierarchy of prepotency as well as behavioral motivation is innately coded. Therefore, physiological needs, such as feeding, surpass safety needs⁴⁰. ”.

7) Besides, please check for typos correction. eg. typo in line 146 : alas (?).

Answer: We would like to apologize and we have now carefully edited the manuscript for its clarity and readability. We paid attention to the typos such as the one mentioned by the reviewer (lines 152-154):

“In the wild, food resources are often limited and could ~~alas~~ be located in a dangerous setting such as impending predation”.

For exhaustive details of our rewriting, please find in the following pages the previously submitted main text annotated, in blue, with our modifications and in strikethrough red for deleted parts:

Original submission # COMMSBIO-20-0343

Title

~~The olfactory Grueneberg ganglion drives food choices under threat~~

The Grueneberg ganglion controls odor-driven food choices under threat

Authors

Julien Brechbühl^{1,2}, Aurélie de Vallière^{1,2,†}, Dean Wood^{1,2,†}, Monique Nenniger Tosato¹ and Marie-Christine Broillet^{1,2,*}.

¹ Department of Pharmacology and Toxicology, Faculty of Biology and Medicine, University of Lausanne, Bugnon 27, CH-1011 Lausanne, Switzerland.

² Department of Biomedical Sciences, Faculty of Biology and Medicine, University of Lausanne, Bugnon 27, CH-1011 Lausanne, Switzerland.

* Correspondence to: Marie-Christine Broillet, Department of Biomedical Sciences, Faculty of Biology and Medicine, University of Lausanne, Bugnon 27, CH-1011 Lausanne, Switzerland. **Email:** mbroille@unil.ch

ORCID to Marie-Christine Broillet: 0000-0002-0487-6638.

ORCID to Julien Brechbühl: 0000-0002-2335-3058.

† These authors contributed equally.

Abstract

The ability to efficiently search for food is fundamental for animal survival. Olfactory messages are used to find food while being aware of the impending risk of predation. How these different olfactory clues are combined to optimize decision-making concerning food selection remains elusive. Here, we find that chemical danger cues drive the food selection in mice via the activation of a specific olfactory subsystem, the Grueneberg ganglion (GG). We show that a functional GG is required to decipher the threatening quality of an unfamiliar food. We also find that the increase in corticosterone, which is GG-dependent, **dynamizes enhances** safe food preference acquired during social transmission. Moreover, we demonstrate that ~~the memorization of a food preference can be rectified~~ **memory retrieval for food preference can be extinguished** by activation of the GG circuitry. Our findings reveal a key function played by the GG in controlling contextual food responses and illustrate how mammalian organisms integrate environmental chemical stress to optimize decision-making.

Introduction

Animals continuously assess the environmental risks and, for survival, may need to compromise between the gain of feeding and the potential cost of exposure to danger¹. Especially, when confronted to food scarcity, behavioral trade-offs need to be made^{2,3}. Olfactory senses play an important role in this process as they help to find food but also to sense and avoid threatening situations⁴⁻⁹ such as predation. For rodents, the attraction to food is primarily determined by familiarity, as scents of known food are innately favored^{2,10}. These scents are detected by olfactory sensory neurons (OSNs) found in the main olfactory epithelium (MOE; Fig. 1a)^{11,12}. Food familiarity can be acquired via communication between individuals, by the so-called “social transmission of food preference” (STFP), a behavioral process that occurs by sniffing the breath of conspecifics and making an association between the novel food scent and the presence of the endogenously-produced carbon disulfide gas (CS₂)^{4,13,14}. For this process to occur, the concomitant recognition of the food odorant by OSNs and the detection of the CS₂ by guanylyl cyclase-D neurons (GC-D; Fig. 1a) of the MOE is essential. On the other hand, odors that signal an impending danger are involuntarily left behind by predators^{15,16} or emitted by stressed congeners as warning alarm pheromones^{9,17}. We showed previously that these danger and warning cues are detected in

a specialized olfactory subsystem, the Grueneberg ganglion^{9,15} (GG; Fig. 1a). They generate, in the recipient mouse, stereotypical fear-related behaviors, such as freezing and risk-assessment as well as stress-related systemic responses^{15,18}.

In this study, we provide evidence and new biological insights on how mice exploit threatening scents to optimize and control their food selection. Using gene-targeted, surgically treated and sham-operated mice as well as a series of integrative behavioral assays, we functionally determine that the GG olfactory subsystem controls innate and acquired food preferences when mice smell an impending danger. The GG is indeed required to decipher the threatening quality of an unfamiliar food. We also find that its activation increases the systemic corticosterone level which in turn **dynamizes** **enhances** safe food preferences acquired during social transmission. Moreover, we finally demonstrate that the acquisition of a food preference itself could be circumstantially **rectified** **reset** by the activation of the GG circuitry, providing a **survival** **decision-making** advantage for mice.

Results

Selective deletion of the GG circuitry. The conflicting environmental olfactory messages carried by **odorants emitted by** familiar food **odorants** and by olfactory danger cues need to be continuously evaluated for risk taking and final food decision-making. To address the functional relevance of the GG in this last process in mice, we first surgically disconnected it by nerve axotomy⁹ (axo; Fig. 1a). Thanks to the expression of the olfactory marker protein (OMP; Fig. 1b and c), a neuronal marker for both the GG and the MOE, we then verified that the axotomy induced specific loss of the GG in axotomized (Axo) but not in control sham-operated mice (Ctrl) (Fig. 1b) without affecting the MOE (Fig. 1c). We next demonstrated that the GC-D neurons were still present in the MOE after GG axotomy (Fig. 1c) thanks to the expression of the enzyme phosphodiesterase 2A (PDE2A; Fig. 1b and c) which is shared by GG and GC-D circuitries¹⁹. We then used transgenic guanylyl cyclase-G (GCG, a marker of GG circuitry)–Cre–green fluorescent protein (GFP) mice¹⁹ to selectively trace the GG connections into their olfactory bulb target (OB; Fig. 1a), the necklace glomerular complex (NG; Fig. 1a). Thereby, we distinctly differentiated, in the NG, the parallel connections between GG and GC-D circuitries and we found **interglomerular cells expressing the tyrosine hydroxylase³⁰ (TH; Fig. 1d and Supplementary Fig. 1a) suggesting potential cross-**

~~interactions. Nevertheless, we next observed, in GCG-Cre-GFP Axo mice,~~ that the GC-D circuitry and, in particular its associated necklace glomeruli, were still intact **in GCG-Cre-GFP Axo mice** (Fig. 4e 1d and Supplementary Fig. 4b 1; Ctrl, $N_{\text{mouse}} = 3$; Axo, $N_{\text{mouse}} = 5$). Moreover, this GC-D circuitry was still active in the presence of CS₂, **independently from the GG axotomy procedure,** as verified by immediate early gene c-Fos expression¹⁵ (Fig. 4f 1e and f; ~~Ctrl, $N_{\text{mouse}} = 3$; Axo, $N_{\text{mouse}} = 5$~~ **Ctrl, $N_{\text{glomeruli}} = 6$; Axo, $N_{\text{glomeruli}} = 7$; one-tailed t -test: $p = 0.272$, ns**). We have then systematically used the selective deletion of this olfactory subsystem to functionally dissect the relevance of the GG circuitry in **odor-driven** food selection when mice were exposed to chemical danger cues.

Cracking the threatening code of food by GG detection.

GG encodes threatening scents and dominates over MOE-transmitted odor signals.

Mice naturally disregard unfamiliar food based on the unknown odorants it releases, moreover familiar food soiled with danger cues is likewise avoided². As a first physiological assay, we therefore tested odorants commonly used to odorize food such as non-synthetic spices¹³ and we found that these odorants were not directly detected by GG neurons. Indeed, we revealed with calcium imaging experiments performed on acute Fura-2 acetoxymethyl ester (Fura; Fig. 2a) loaded GG slice preparations from transgenic OMP-GFP mice⁹ that, in a total of 54 living GFP tagged GG neurons ($N_{\text{mouse}} = 4$; $n_{\text{slice}} = 6$), Cinnamon, Cocoa, Anise, Oregano, Thyme, Basil, Nutmeg, Ginger as well as the standard mice food and CS₂ did not generate any GG neuronal activity (Fig. 2b). On the other hand, the predator-derived cues 2-propylthietane (2PT) from the stoat anal glands, the mouse alarm pheromone 2-sec-butyl-4,5-dihydrothiazole (SBT) as well as mountain lion (Mt.Lion) urine^{15,20} initiated **strong** reversible calcium responses in respectively 83 %, 57 % and 100 % of the GG neurons. Thus, this first approach not only suggest that the unfamiliarity of a food encoded by its emitted odorants could not be directly deciphered by the GG but also confirmed the ability of the GG to identify danger cues potentially emitted by soiled food.

We next verified, in an integrative context of food choice, the implication of the GG in decoding food unfamiliarity **based on the odorants emitted**. Ctrl and Axo mice were challenged to select, between familiar and unfamiliar food in a two choices assay (Fig. 2c and Supplementary Fig. 2a). To that purpose,

two familiar powdered foods were proposed to mice, odorized either with a series of never-encountered before odorants (odor #1; as unfamiliar food) or with the odorless water (odor #2; as familiar food). We placed all the tested odorants around the powdered foods to exclude any toxicity potentially displayed by synthetic cues. Subsequently, the preference for an odorized food was calculated as the ratio between the consumption of food odor #1 versus the total food consumed (food odor #1 + odor #2) in which the 0.5 value, corresponding to the non-preference threshold was subtracted. Values were thus expressed between 0.5 to -0.5 where positive scores corresponded to a preference for the food odor #1; negative scores for the food odor #2 and zero corresponded to no preference displayed. Hence we showed that rodents indeed prefer familiar food² as non-synthetic spices such as the unfamiliar Cinnamon or Cocoa¹³ as odor #1 were not preferred by mice. Moreover, we observed that this innate choice was also performed by Axo mice (Fig. 2d), confirming that this avoidance behavior was indeed not directly dependent of the GG detection (Fig. 2a-b) but also pointed out to the conserved MOE functionality in the GG axotomized mouse model. Then we found that the GC-D-related ligand CS₂, without any social context¹⁰, did not influence diet selection in both phenotypes (Fig. 2d). We next tested butyric acid (BA) a known aversive odorant that smells rancid and has no alerting relevance¹⁵ and found that it indeed generated **strong** food avoidance both in Ctrl and Axo mice confirming its previously reported GG-independent detection¹⁵ (Fig. 2d). Food odorized with predator scents were also **strongly** aversive (Fig. 2d) as observed with the pyrazine analogues (Pyrazines), found for example in Mt.Lion urine²⁰, 2-PT, 2,4,5-trimethylthiazoline (TMT) from the red fox feces as well as the mouse alarm pheromone SBT. Nevertheless, in Axo mice, this dislike was significantly reduced indicating and confirming (Fig. 2b) that the GG was indeed implicated in the perception of these chemical danger cues¹⁵. Finally, we used Mt.Lion urine as a natural source of predator scents²⁰ and found that its aversive effect on food choice was exclusively dependent on a functional GG (Fig. 2d). Thus, confirming our GG calcium imaging investigations (Fig. 2a-b) that show that mice do not recognize **aversive-caution** odor signals emitted by unfamiliar food via GG detection as demonstrated by exposing them to familiar versus unfamiliar foods (Fig. 2d).

We next highlighted that this olfactory subsystem was fundamental for mice to decipher the threatening quality of an unfamiliar food (Fig. 2e). Indeed, on sets of naïve mice, when we used the pungent and unfamiliar BA as odor #2 (Fig. 2e) in the previous two choices assay (Fig. 2c), mice from

both phenotypes now preferred Cinnamon, Cocoa and CS₂ as sources of unfamiliar odorized food (Fig. 2e) ~~demonstrating the robust~~ confirming the previously observed aversity of the BA (Fig. 2c). Interestingly, ~~this behavior was entirely reversed~~ in the presence of danger cues such as the Pyrazines, 2PT, TMT, SBT and Mt.Lion urine, ~~which were strongly avoided by~~ Ctrl mice ~~that~~ now systematically preferred the aversive and unfamiliar BA (Fig. 2e). TMT and SBT were particularly efficient as they were still able to generate this innate reaction in serial dilutions (Supplementary Fig. 2b and c). Remarkably, this observed ~~innate food selection was impaired~~ preference for BA disappeared in the absence of a functional GG, without affecting the total food consumption (Supplementary Fig. 2d and e). Thus, mice decode the threatening quality of unfamiliar food by GG detection.

Taken together, our results show that, the GG acts as an immediate ~~and essential~~ sensor, which deciphers the threatening quality of odorants emitted by food. Mice will take their final consumption decision about the safety of the resource thanks to its smell and to its GG perception.

Threatening scents activate GG-dependent corticosterone responses. In the wild, food resources are often limited and could ~~alas~~ be located in a dangerous setting such as impending predation. Nevertheless, motivational state is modified under hunger context³. To evaluate trade-offs displayed by fasting mice when confronted to actively searching a food resource in a danger context, we next challenged Ctrl and Axo mice with unreachable food resources moistened with SBT and with Mt.Lion urine, as sources of respectively intra- and inter-species conditioning stimuli (+ C.S) mimicking environmental evidences of olfactory threats (Fig. 3a). Remarkably, we observed that mice displayed risk assessment behavior (Fig. 3a) in both phenotypes¹⁵, indicating that, in a context of food scarceness, the opportunity to eat indeed overrules fear (+ Water: 8.2 ± 1.0 per min; + SBT: 7.2 ± 1.3 per min; + Mt.Lion: 7.6 ± 1.0 per min. $N_{\text{mouse}} = 8$. RM one-way ANOVA: $F_{(2,14)} = 0.9016$, $p = 0.428$, ns). This behavior ~~also highlights that odorants emitted by food were sufficient to initiate this motivational-related behavioral process independently from GG detection~~ (Ctrl, $N_{\text{mouse}} = 12$; Axo, $N_{\text{mouse}} = 12$; one-tailed w -test: $p = 0.483$, ns). We then followed the systemic integration of these stressful situations with the expression of the immediate early gene c-Fos²¹. We focused on the amygdalopiriform transition area (APir; Fig. 3b) which is the brain region implicated in the increase of stress-related hormone level in the blood when mice smell volatile predator scents^{15,22}. We

found that this brain region was significantly activated by both SBT and the predator urine (Fig. 3c-d). We next confirmed this result by measuring an increase of the systemic corticosterone level (Fig. 3e), implying that intra- and inter-species danger cues were both processed in this specific APir nucleus. In a dilution series of intraperitoneal corticosterone injections (Cort. i.p.; Fig. 3f), we were further able to mimic the observed stress-related hormonal elevation with an amount of Cort. i.p. of 5.0 mg kg⁻¹, **therefore bypassing APir activation** in absence of conditioning stimuli (- C.S; Fig. 3f **and Supplementary Fig. 3**). Moreover, we found that, in Axo mice, the activation of the APir region as well as the elevation of the systemic corticosterone were significantly impaired in this threatening context (Fig. 3c to e), demonstrating that, although the GG was not involved **in the risk-taking of mice** in **odor-driven** foraging, it was essential for the hormonal and physiological adaptation to danger sensing.

Elevation of corticosterone optimizes learned-food selection. In the search of food resources, knowledge of familiar food obtained by STFP support the food decision-making by a recall memory process that requires the activation of a specific part of the hippocampus, the ventral subiculum^{13,23} (VS; Fig. 3b). In this study, we found that the presence of danger cues enhanced this learned food-**odorant** preference when we performed a two choices assay under threatening conditions. For that, **we first trained mice to develop a food-odorant preference. Briefly**, a demonstrator mouse **first** ate an odorized demonstrating food (de.food; standard powdered food odorized for example with Cinnamon as a spice #1; Phase 1; Fig. 4a). Then it returned with its observer littermates to allow the STFP learning process to happen (Phase 2; Fig. 4a). Finally, after 24 hours of fasting, the observer mice were individually confronted to a two choices assay between two odorized foods, the demonstrating food and a novel food (new.food; standard powdered food odorized for example with Cocoa as spice #2; Phase 3; Fig. 4b), both surrounded with the same conditioning stimulus (+ C.S; Fig. 4b). To avoid any individual innate preference, spices were used in a counterbalanced mode¹³ (Supplementary Fig. **4a** to c). We observed in both phenotypes that STFP was, indeed, required to develop a food preference (STFP+ | + Water; Fig. 4c) that was, remarkably, not impacted by a chemical confusion generated by the pungent BA (STFP+ | + BA; Fig. 4c). These first effects not only confirmed the conserved functionality of the GC-D circuitry in Axo mice (Fig. 1c **and e-f**) but also pointed out that, **under aversive distraction, mice still olfactively decipher**

food preferences. Surprisingly, we obtained, with the danger cue SBT (STFP+ | + SBT) and with Mt.Lion urine (STFP+ | + Mt.Lion) a significant enhancement of the food-**odorant** preference (Fig. 4c). Indeed, the observed mean food preferences were amplified by around 60 % compared to STFP under standard environmental (STFP+ | + Water) or under pungent conditions (STFP+ | + BA). This enhancement was observed in Ctrl but not in Axo mice. It was found to be both independent from food consumption (Supplementary Fig.-4 **5a**) and from an improvement of the recall memory process as no significant increase in c-Fos activity was observed in the VS brain area under these conditions (Fig. 4d and e). Interestingly, we found that injection of Cort. i.p. 5.0 mg kg⁻¹ under neutral context (STFP+ | Cort. i.p.) was sufficient to mimic this behavioral improvement both in Ctrl and Axo mice (Fig. 4c) **without affecting the memory retrieval activity of the VS (Supplementary Fig. 5b and c)**. We thus demonstrated here that, when danger cues are sensed by the GG, the increase in the systemic corticosterone level that occurs (Fig. 3e) leads to an enhancement of the food preference previously acquired by STFP.

GG circuitry activation selectively erase safety food memory. Conspecific interactions are useful for mice to get food familiarity. We found here that mice can also benefit from threatening chemical information present in their environment to reset this previously acquired food-**odorant** familiarity when associated with a danger context. We indeed tested the impact of olfactory threats on the acquisition of a **new food-odorant preference and thus as we** challenged mice 1 hour after an STFP procedure (Phase 1; Fig. 5a) with a surrogate display that contained the unreachable demonstrating food moistened with an associated conditioning stimulus (Phase 2; + C.S; Fig. 5a). After 24 hours of fasting, the observer mice were then tested in a two choices assay (Phase 3; Fig. 5b). Unexpectedly, we found that, in association with the danger cues SBT (STFP+ / + SBT) or with Mt.Lion urine (STFP+ / + Mt.Lion), Ctrl mice did not display food-**odorant** preferences while they were still observed in Axo mice (Fig. 5c). As a control, we verified that GG-related ligands directly associated with a demonstrating food could not act as conditioning stimuli promoting a food preference or avoidance (Supplementary Fig.-5 **6a to c**), confirming our previous observations that avoidance towards an unfamiliar odorized food is innately coded (Fig. 2d). Besides, we also observed that this apparent and selective amnesia was independent from food consumption (Supplementary Fig.-6 **7a**) or corticosterone elevation. Indeed, mice injected with Cort. i.p. 5.0 mg kg⁻¹

instead of the associated conditioning stimuli (STFP+ / Cort. i.p.) still displayed a food-odorant preference (Fig. 5c). Moreover, we observed a striking resetting of the VS brain area (c-Fos stainings; Fig. 5d and e), that strongly suggests an impairment in the recall memory for a food preference or for its consolidation. Remarkably, as this process was independent from the systemic corticosterone level (Supplementary Fig. 7b and c) and not observed in Axo mice, this cerebral reorientation adaptation was thus directly related with the activation of the GG neuronal circuitry itself (Fig. 5d and e).

Discussion

Acute stress can modulate both the performance of action-outcomes and the learning processes in animals including rodents and humans^{24,25}. Impending danger such as the risk of predation increases stress-related hormonal levels in the blood^{5,15} and can affect memorization²⁶. When faced with imminent danger, alerting senses like the olfactory senses allow the detection of chemical cues signaling for the presence of predators²⁷. The olfactory senses are also essential to find food and therefore trade-offs and sorting out of the environmental chemical information has to take place followed by the appropriate behavioral responses. In this study, we investigated the biological relevance of the detection of chemical danger cues by the GG on odor-driven food selection because of its fundamental need for animal survival. We demonstrated that the GG circuitry favors innate odor-driven food decision-making in a context of danger (Fig. 2). Indeed, the preys with a functional GG are able to select the food resource, depending on its smell, while avoiding to be eaten by predators. We have also observed that the GG was not directly implicated in the detection of odorants emitted by food resources (Fig. 2) and in the linked foraging behavior (Fig. 3), we can thus consider that our GG-dependent results are not exclusively specific to food selection^{15,17} and that they may be extended to the global effects of threatening scents on odor-driven preferences.

Our understanding of the interplay between the pathways processing GG-induced fear-like responses and those encoding food-odorant information remains to be completed. GG neurons use multi-signaling pathways, principally linked to bitter taste signaling²⁸ and guanylyl cyclase-G cascades¹⁸ to detect a large repertoire of chemical danger cues^{28,29}. Interestingly, in GC-D neurons, a similar encoding strategy occurs¹⁴ making it also sensitive to fatty acids, steroid hormones as well as to some GG-related

cues such as pyrazines. This shared encoding strategy as well as the partial overlap of recognized cues between the GC-D and the GG subsystems might, at the detection level, contribute to the apparent positive collaboration between these two parallel circuitries.

In a pilot experiment performed in the NG complex of GCG–Cre–GFP mice, we noticed the presence of periglomerular cells expressing the tyrosine hydroxylase^{30,31} (TH; Supplementary Fig. 8a and b). Interestingly, their density was preferentially located around GC-D glomeruli (GC-D: 9.3 ± 0.9 TH+ cells, $N_{glomeruli} = 6$; GG: 5.0 ± 1.2 TH+ cells, $N_{glomeruli} = 7$), supporting the interconnectivity of GC-D glomeruli with other olfactory circuitries such as those of the MOE³². On the other hand and compared to other OB regions³³, TH+ cell density was moderate around GG glomeruli, which is consistent with their homogenous innervation¹⁹ and their episodic afferent activity⁹. This might question the potential cross-glomerular regulatory task of these periglomerular cells. Moreover, and as a temporal hiatus, we found that the GG circuitry could impact socially learned food preferences in a delayed manner (Fig. 4 and Fig.5), at least 1 hour after the STFP procedure (Fig. 4). Therefore periglomerular regulation emerging from GG on GC-D glomeruli requires further investigations.

The integration of the GG circuitry on higher structures in the CNS have revealed specific subregions implicated in fear and anxiety responses such as the posteroventral division of the medial amygdala (MeA) and the dorsomedial subdivision of the ventromedial hypothalamus^{16,34} (VMH). Our findings unravel two additional brain domains implicated in this global integration, the APir (Fig. 3) and the VS (Fig. 4 and Fig. 5). Interestingly, the APir was previously reported as a specific area of the olfactory cortex that induces stress hormone responses to volatile predator scents²¹. Here, we demonstrate that this heterospecific response could be extended to conspecific-signaling integration, as the mouse alarm pheromone SBT likewise initiates APir activities (Fig. 3). In addition to the instinctive fear response, we therefore uncovered here a key consequence of the subsequent systemic corticosterone elevation and demonstrated its central outcome for the improvement of odor-driven food behavior (Fig. 4). The processing ability of the CNS structures, such as the one observed here with the APir is usually dependent of their activation. On the other hand, we found that the treatment of the information by the VS emerging from the GG circuitry consisted in its ability to reset its memory-retrieval activity for socially-learned food response^{13,23} (Fig. 5). This last finding is consistent with previous reports implying

threatening scents in the impairment of retrieval memory³⁵ but also supports the VS as a key CNS area implicated in context-dependent stress and integration of fear memories³⁶⁻³⁸. Moreover, we have observed that the same chemical danger cues can act both during (Fig. 4) and upstream (Fig. 5) of the decision-making event with opposing behavioral outcomes. Threatening scents could thus develop contrasting effects on odor-driven food choices in which the VS should play a context-dependent integrative role. These observations further imply a complex strategy for encoding both the predator-related stress as well as the retrieval of innate and learned fear³⁹. At the individual level, this innate trait could be considered as a beneficial mechanism of protection. Indeed, when the food choice is presented in a context of threat (Fig. 4), the favored decision for survival should be to select a safe and familiar food. Nonetheless, to be efficient and to decrease the risk of being eaten, a compromise is made and the emerging strategy is, in the end, to erase this food familiarity and to consume the food anyway (Fig. 3 and Fig. 5). Thus, for mice, as for humans, hierarchy of prepotency as well as behavioral motivation is innately coded. Therefore, physiological needs, such as feeding, surpass safety needs⁴⁰.

In the wild, the ability to take rapid decisions concerning food choice is critical for the survival of the individual and leads to an increase in the overall fitness of the species. It is therefore conserved throughout the animal kingdom^{41,42}. For example, in the nematode *Caenorhabditis elegans* (*C. elegans*), stress linked to environmental conditions can impact food preferences when they are coupled with odorants⁴³ as well as with alarm pheromones and predator scents^{44,45}. Interestingly, this worm modality is dependent on a specific olfactory subsystem composed of a pair of amphid wing cells of type C⁴⁶ (AWC). GG neurons were reported to share molecular and functional features with these AWC neurons, conferring to these two olfactory subsystems an inter-species orthologous status²⁹.

In summary, our present study demonstrates that the GG potentiates, in mice, the decision-making process for the odor-driven food choice under threat. Our results highlight how mice integrate environmental stressful stimuli to control both innate and previously acquired odor-driven food preferences. Olfactory threats are detected by the GG that operates as an immediate sensor acting both on innate perceptions, systemic hormonal levels and on the acquired food preferences to circumstantially revise mice decision-making, a mechanism that confers to the animal a beneficial survival advantage²¹.

Responses to reviewers:

COMMSBIO-20-0343 (The olfactory Grueneberg ganglion drives food choices under threat).

Reviewer #3:

We would like to thank the reviewer for her/his comments. We truly appreciate her/his insightful review of our manuscript. We have now addressed all of her/his questions and suggestions and we believe that taking them into account significantly strengthen our revised manuscript.

In summary, in order to take into account all the comments from the three reviewers, we have now modified the manuscript as follows: We have described the observed effects of threatening scents on *odor-driven food choices* rather than on *food preference*. We have now performed additional control experiments to complete and strengthen our previous observations and accordingly, we have redesigned the Figure 1 and the Figure 3a. We created new Supplementary figures 3 and 8. We also redesigned Supplementary figure 1 and the original Supplementary figures 4 and 6 (our new Supplementary figures 5 and 7). We have now detailed the meaning of the preference ratio in the main text. As suggested by the reviewers we have modified our discussion section to include the important and previously missing notions on the global integration of the GG circuitry (detection, bulbar / CNS integration and output; adding also new references). We have also addressed the contrasting findings of Figures 4 and 5. We have now carefully edited the manuscript for typos and for its clarity and readability.

We hope that these changes will allow a better understanding and interpretation of our results and we thus propose to also modify the title of our manuscript accordingly. We look forward to the reviewer response and we will be glad to respond to any further questions and comments that she/he may have.

Please find here the responses to her/his specific comments:

1) My main concern is the presentation of the results. The writing is often unclear, and sometimes misleading, a very smooth writing style notwithstanding. The discussion of the somewhat opposing results of figure 4 and 5 could be improved.

Answer: We would like to thank the reviewer for raising this important concern and apologize for the lack of clarity regarding the presentation of our results. We have now paid more attention to our description in the main text by avoiding any misleading terminology. **For exhaustive details of our rewriting, please find at the end of our Responses the previously submitted text annotated, in blue, with our modifications and in strikethrough red for deleted parts.**

We have also considered the comments from all reviewers and significantly changed the overall structure of the discussion. Please find below the specific comments that correspond to the mentioned opposing results of Figure 4 and 5 (*lines 275-303*):

“The integration of the GG circuitry on higher structures in the CNS have revealed specific subregions implicated in fear and anxiety responses such as the posteroventral division of the medial amygdala (MeA) and the dorsomedial subdivision of the ventromedial hypothalamus^{16,34} (VMH). Our findings unravel two additional brain domains implicated in this global integration, the APir (Fig. 3) and the VS (Fig. 4 and Fig. 5). Interestingly, the APir was previously reported as a specific area of the olfactory cortex that induces stress hormone responses to volatile predator scents²¹. Here, we demonstrate that this heterospecific response could be extended to conspecific-signaling integration, as the mouse alarm pheromone SBT likewise initiates APir activities (Fig. 3). In addition to the instinctive fear response, we therefore uncovered here a key consequence of the subsequent systemic corticosterone elevation and demonstrated its central outcome for the improvement of odor-driven food behavior (Fig. 4). The processing ability of the CNS structures, such as the one observed here with the APir is usually dependent of their activation. On the other hand, we found that the treatment of the information by the VS emerging from the GG circuitry consisted in its ability to reset its memory-retrieval activity for socially-learned food response^{13,23} (Fig. 5). This last finding is consistent with previous reports implying threatening scents in the impairment of retrieval memory³⁵ but also supports the VS as a key CNS area implicated in context-dependent stress and integration of fear

memories³⁶⁻³⁸. Moreover, we have observed that the same chemical danger cues can act both during (Fig. 4) and upstream (Fig. 5) of the decision-making event with opposing behavioral outcomes. Threatening scents could thus develop contrasting effects on odor-driven food choices in which the VS should play a context-dependent integrative role. These observations further imply a complex strategy for encoding both the predator-related stress as well as the retrieval of innate and learned fear³⁹. At the individual level, this innate trait could be considered as a beneficial mechanism of protection. Indeed, when the food choice is presented in a context of threat (Fig. 4), the favored decision for survival should be to select a safe and familiar food. Nonetheless, to be efficient and to decrease the risk of being eaten, a compromise is made and the emerging strategy is, in the end, to erase this food familiarity and to consume the food anyway (Fig. 3 and Fig. 5). Thus, for mice, as for humans, hierarchy of prepotency as well as behavioral motivation is innately coded. Therefore, physiological needs, such as feeding, surpass safety needs⁴⁰. ”.

2) Line 30-32, Abstract „We also find that the increase in corticosterone, which is GG-dependent, dynamizes safe food preference acquired during social transmission. Moreover, we demonstrate that the memorization of a food preference can be rectified by activation of the GG circuitry.,,

I do not think these sentences are understandable. As far as I can tell, the authors use 'dynamize' (only used in Abstract and Introduction, not in Results) to describe a more pronounced preference for safe (socially accepted) food, when GG is simultaneously activated by predator odor. Maybe use 'enhance'?

How can a memory be rectified? Again the term is not used in Results, only in Abstract and Introduction. The reference given for 'memorization' deals with memory retrieval.

Answer: We agree with the reviewer that we used a too descriptive and therefore inadequate terminology and we apologize for the lack of systematic uses of words between the original abstract, introduction and result sections. Indeed, the use of “memorization of a food preference can be rectified” was a misleading notion and we have now modified the abstract and the main text. **For exhaustive details of our rewriting, please find at the end of our Responses**

the previously submitted text annotated, in blue, with our modifications and in strikethrough red for deleted parts.

Please find here our new abstract annotated with our modifications (*lines 25-35*):

*“The ability to efficiently search for food is fundamental for animal survival. Olfactory messages are used to find food while being aware of the impending risk of predation. How these different olfactory clues are combined to optimize decision-making concerning food selection remains elusive. Here, we find that chemical danger cues drive the food selection in mice via the activation of a specific olfactory subsystem, the Grueneberg ganglion (GG). We show that a functional GG is required to decipher the threatening quality of an unfamiliar food. We also find that the increase in corticosterone, which is GG-dependent, ~~dynamizes~~ **enhances** safe food preference acquired during social transmission. Moreover, we demonstrate that ~~the memorization of a food preference can be rectified~~ **memory retrieval for food preference can be extinguished** by activation of the GG circuitry. Our findings reveal a key function played by the GG in controlling contextual food responses and illustrate how mammalian organisms integrate environmental chemical stress to optimize decision-making.”.*

Please find here our new text annotated with our modifications (*lines 60-64*):

*“We also find that its activation increases the systemic corticosterone level which in turn ~~dynamizes~~ **enhances** safe food preferences acquired during social transmission. Moreover, we finally demonstrate that the acquisition of a food preference itself could be circumstantially ~~rectified~~ **reset** by the activation of the GG circuitry, providing a ~~survival~~ **decision-making** advantage for mice”.*

3) Line 89 Cracking the threatening code of food by GG detection.

Sounds impressive but is not very clear. Maybe it would be better to write something like „GG encodes predator odors and dominates over MOE-transmitted odor signals“.

Answer: As suggested by the reviewer, we have now modified this subtitle (*lines 89*):

~~“Cracking the threatening code of food by GG detection. “GG encodes threatening scents and dominates over MOE-transmitted odor signals”~~”.

4) *Line 250-253*

I do not see how the speculation mentioned follows from the results reported here.

Answer: We agree with the reviewer that this argument was speculative. We have now deleted the sentence ~~“we can therefore speculate that the GG is an olfactory subsystem adding survival advantages against the environmental pressure and thus has been conserved throughout evolution”~~ from our new discussion.

For exhaustive details of our rewriting, please find in the following pages the previously submitted main text annotated, in blue, with our modifications and in strikethrough red for deleted parts:

Original submission # COMMSBIO-20-0343

Title

~~The olfactory Grueneberg ganglion drives food choices under threat~~

The Grueneberg ganglion controls odor-driven food choices under threat

Authors

Julien Brechbühl^{1,2}, Aurélie de Vallière^{1,2,†}, Dean Wood^{1,2,†}, Monique Nenniger Tosato¹ and Marie-Christine Broillet^{1,2,*}.

¹ Department of Pharmacology and Toxicology, Faculty of Biology and Medicine, University of Lausanne, Bugnon 27, CH-1011 Lausanne, Switzerland.

² Department of Biomedical Sciences, Faculty of Biology and Medicine, University of Lausanne, Bugnon 27, CH-1011 Lausanne, Switzerland.

* Correspondence to: Marie-Christine Broillet, Department of Biomedical Sciences, Faculty of Biology and Medicine, University of Lausanne, Bugnon 27, CH-1011 Lausanne, Switzerland. **Email:** mbroille@unil.ch

ORCID to Marie-Christine Broillet: 0000-0002-0487-6638.

ORCID to Julien Brechbühl: 0000-0002-2335-3058.

† These authors contributed equally.

Abstract

The ability to efficiently search for food is fundamental for animal survival. Olfactory messages are used to find food while being aware of the impending risk of predation. How these different olfactory clues are combined to optimize decision-making concerning food selection remains elusive. Here, we find that chemical danger cues drive the food selection in mice via the activation of a specific olfactory subsystem, the Grueneberg ganglion (GG). We show that a functional GG is required to decipher the threatening quality of an unfamiliar food. We also find that the increase in corticosterone, which is GG-dependent, **dynamizes enhances** safe food preference acquired during social transmission. Moreover, we demonstrate that ~~the memorization of a food preference can be rectified~~ **memory retrieval for food preference can be extinguished** by activation of the GG circuitry. Our findings reveal a key function played by the GG in controlling contextual food responses and illustrate how mammalian organisms integrate environmental chemical stress to optimize decision-making.

Introduction

Animals continuously assess the environmental risks and, for survival, may need to compromise between the gain of feeding and the potential cost of exposure to danger¹. Especially, when confronted to food scarceness, behavioral trade-offs need to be made^{2,3}. Olfactory senses play an important role in this process as they help to find food but also to sense and avoid threatening situations⁴⁻⁹ such as predation. For rodents, the attraction to food is primarily determined by familiarity, as scents of known food are innately favored^{2,10}. These scents are detected by olfactory sensory neurons (OSNs) found in the main olfactory epithelium (MOE; Fig. 1a)^{11,12}. Food familiarity can be acquired via communication between individuals, by the so-called “social transmission of food preference” (STFP), a behavioral process that occurs by sniffing the breath of conspecifics and making an association between the novel food scent and the presence of the endogenously-produced carbon disulfide gas (CS₂)^{4,13,14}. For this process to occur, the concomitant recognition of the food odorant by OSNs and the detection of the CS₂ by guanylyl cyclase-D neurons (GC-D; Fig. 1a) of the MOE is essential. On the other hand, odors that signal an impending danger are involuntarily left behind by predators^{15,16} or emitted by stressed congeners as warning alarm pheromones^{9,17}. We showed previously that these danger and warning cues are detected in

a specialized olfactory subsystem, the Grueneberg ganglion^{9,15} (GG; Fig. 1a). They generate, in the recipient mouse, stereotypical fear-related behaviors, such as freezing and risk-assessment as well as stress-related systemic responses^{15,18}.

In this study, we provide evidence and new biological insights on how mice exploit threatening scents to optimize and control their food selection. Using gene-targeted, surgically treated and sham-operated mice as well as a series of integrative behavioral assays, we functionally determine that the GG olfactory subsystem controls innate and acquired food preferences when mice smell an impending danger. The GG is indeed required to decipher the threatening quality of an unfamiliar food. We also find that its activation increases the systemic corticosterone level which in turn **dynamizes** **enhances** safe food preferences acquired during social transmission. Moreover, we finally demonstrate that the acquisition of a food preference itself could be circumstantially **rectified** **reset** by the activation of the GG circuitry, providing a **survival** **decision-making** advantage for mice.

Results

Selective deletion of the GG circuitry. The conflicting environmental olfactory messages carried by **odorants emitted by** familiar food **odorants** and by olfactory danger cues need to be continuously evaluated for risk taking and final food decision-making. To address the functional relevance of the GG in this last process in mice, we first surgically disconnected it by nerve axotomy⁹ (axo; Fig. 1a). Thanks to the expression of the olfactory marker protein (OMP; Fig. 1b and c), a neuronal marker for both the GG and the MOE, we then verified that the axotomy induced specific loss of the GG in axotomized (Axo) but not in control sham-operated mice (Ctrl) (Fig. 1b) without affecting the MOE (Fig. 1c). We next demonstrated that the GC-D neurons were still present in the MOE after GG axotomy (Fig. 1c) thanks to the expression of the enzyme phosphodiesterase 2A (PDE2A; Fig. 1b and c) which is shared by GG and GC-D circuitries¹⁹. We then used transgenic guanylyl cyclase-G (GCG, a marker of GG circuitry)–Cre–green fluorescent protein (GFP) mice¹⁹ to selectively trace the GG connections into their olfactory bulb target (OB; Fig. 1a), the necklace glomerular complex (NG; Fig. 1a). Thereby, we distinctly differentiated, in the NG, the parallel connections between GG and GC-D circuitries and we found **interglomerular cells expressing the tyrosine hydroxylase³⁰ (TH; Fig. 1d and Supplementary Fig. 1a) suggesting potential cross-**

~~interactions. Nevertheless, we next observed, in GCG-Cre-GFP Axo mice,~~ that the GC-D circuitry and, in particular its associated necklace glomeruli, were still intact **in GCG-Cre-GFP Axo mice** (Fig. 4e 1d and Supplementary Fig. 4b 1; Ctrl, $N_{\text{mouse}} = 3$; Axo, $N_{\text{mouse}} = 5$). Moreover, this GC-D circuitry was still active in the presence of CS₂, **independently from the GG axotomy procedure**, as verified by immediate early gene c-Fos expression¹⁵ (Fig. 4f 1e and f; ~~Ctrl, $N_{\text{mouse}} = 3$; Axo, $N_{\text{mouse}} = 5$~~ **Ctrl, $N_{\text{glomeruli}} = 6$; Axo, $N_{\text{glomeruli}} = 7$; one-tailed t -test: $p = 0.272$, ns**). We have then systematically used the selective deletion of this olfactory subsystem to functionally dissect the relevance of the GG circuitry in **odor-driven** food selection when mice were exposed to chemical danger cues.

Cracking the threatening code of food by GG detection.

GG encodes threatening scents and dominates over MOE-transmitted odor signals.

Mice naturally disregard unfamiliar food based on the unknown odorants it releases, moreover familiar food soiled with danger cues is likewise avoided². As a first physiological assay, we therefore tested odorants commonly used to odorize food such as non-synthetic spices¹³ and we found that these odorants were not directly detected by GG neurons. Indeed, we revealed with calcium imaging experiments performed on acute Fura-2 acetoxymethyl ester (Fura; Fig. 2a) loaded GG slice preparations from transgenic OMP-GFP mice⁹ that, in a total of 54 living GFP tagged GG neurons ($N_{\text{mouse}} = 4$; $n_{\text{slice}} = 6$), Cinnamon, Cocoa, Anise, Oregano, Thyme, Basil, Nutmeg, Ginger as well as the standard mice food and CS₂ did not generate any GG neuronal activity (Fig. 2b). On the other hand, the predator-derived cues 2-propylthietane (2PT) from the stoat anal glands, the mouse alarm pheromone 2-sec-butyl-4,5-dihydrothiazole (SBT) as well as mountain lion (Mt.Lion) urine^{15,20} initiated **strong** reversible calcium responses in respectively 83 %, 57 % and 100 % of the GG neurons. Thus, this first approach not only suggest that the unfamiliarity of a food encoded by its emitted odorants could not be directly deciphered by the GG but also confirmed the ability of the GG to identify danger cues potentially emitted by soiled food.

We next verified, in an integrative context of food choice, the implication of the GG in decoding food unfamiliarity **based on the odorants emitted**. Ctrl and Axo mice were challenged to select, between familiar and unfamiliar food in a two choices assay (Fig. 2c and Supplementary Fig. 2a). To that purpose,

two familiar powdered foods were proposed to mice, odorized either with a series of never-encountered before odorants (odor #1; as unfamiliar food) or with the odorless water (odor #2; as familiar food). We placed all the tested odorants around the powdered foods to exclude any toxicity potentially displayed by synthetic cues. Subsequently, the preference for an odorized food was calculated as the ratio between the consumption of food odor #1 versus the total food consumed (food odor #1 + odor #2) in which the 0.5 value, corresponding to the non-preference threshold was subtracted. Values were thus expressed between 0.5 to -0.5 where positive scores corresponded to a preference for the food odor #1; negative scores for the food odor #2 and zero corresponded to no preference displayed. Hence we showed that rodents indeed prefer familiar food² as non-synthetic spices such as the unfamiliar Cinnamon or Cocoa¹³ as odor #1 were not preferred by mice. Moreover, we observed that this innate choice was also performed by Axo mice (Fig. 2d), confirming that this avoidance behavior was indeed not directly dependent of the GG detection (Fig. 2a-b) but also pointed out to the conserved MOE functionality in the GG axotomized mouse model. Then we found that the GC-D-related ligand CS₂, without any social context¹⁰, did not influence diet selection in both phenotypes (Fig. 2d). We next tested butyric acid (BA) a known aversive odorant that smells rancid and has no alerting relevance¹⁵ and found that it indeed generated **strong** food avoidance both in Ctrl and Axo mice confirming its previously reported GG-independent detection¹⁵ (Fig. 2d). Food odorized with predator scents were also **strongly** aversive (Fig. 2d) as observed with the pyrazine analogues (Pyrazines), found for example in Mt.Lion urine²⁰, 2-PT, 2,4,5-trimethylthiazoline (TMT) from the red fox feces as well as the mouse alarm pheromone SBT. Nevertheless, in Axo mice, this dislike was significantly reduced indicating and confirming (Fig. 2b) that the GG was indeed implicated in the perception of these chemical danger cues¹⁵. Finally, we used Mt.Lion urine as a natural source of predator scents²⁰ and found that its aversive effect on food choice was exclusively dependent on a functional GG (Fig. 2d). Thus, confirming our GG calcium imaging investigations (Fig. 2a-b) that show that mice do not recognize **aversive-caution** odor signals emitted by unfamiliar food via GG detection as demonstrated by exposing them to familiar versus unfamiliar foods (Fig. 2d).

We next highlighted that this olfactory subsystem was fundamental for mice to decipher the threatening quality of an unfamiliar food (Fig. 2e). Indeed, on sets of naïve mice, when we used the pungent and unfamiliar BA as odor #2 (Fig. 2e) in the previous two choices assay (Fig. 2c), mice from

both phenotypes now preferred Cinnamon, Cocoa and CS₂ as sources of unfamiliar odorized food (Fig. 2e) ~~demonstrating the robust~~ confirming the previously observed aversity of the BA (Fig. 2c). Interestingly, ~~this behavior was entirely reversed~~ in the presence of danger cues such as the Pyrazines, 2PT, TMT, SBT and Mt.Lion urine, ~~which were strongly avoided by~~ Ctrl mice ~~that~~ now systematically preferred the aversive and unfamiliar BA (Fig. 2e). TMT and SBT were particularly efficient as they were still able to generate this innate reaction in serial dilutions (Supplementary Fig. 2b and c). Remarkably, this observed ~~innate food selection was impaired~~ preference for BA disappeared in the absence of a functional GG, without affecting the total food consumption (Supplementary Fig. 2d and e). Thus, mice decode the threatening quality of unfamiliar food by GG detection.

Taken together, our results show that, the GG acts as an immediate ~~and essential~~ sensor, which deciphers the threatening quality of odorants emitted by food. Mice will take their final consumption decision about the safety of the resource thanks to its smell and to its GG perception.

Threatening scents activate GG-dependent corticosterone responses. In the wild, food resources are often limited and could ~~alas~~ be located in a dangerous setting such as impending predation. Nevertheless, motivational state is modified under hunger context³. To evaluate trade-offs displayed by fasting mice when confronted to actively searching a food resource in a danger context, we next challenged Ctrl and Axo mice with unreachable food resources moistened with SBT and with Mt.Lion urine, as sources of respectively intra- and inter-species conditioning stimuli (+ C.S) mimicking environmental evidences of olfactory threats (Fig. 3a). Remarkably, we observed that mice displayed risk assessment behavior (Fig. 3a) in both phenotypes¹⁵, indicating that, in a context of food scarceness, the opportunity to eat indeed overrules fear (+ Water: 8.2 ± 1.0 per min; + SBT: 7.2 ± 1.3 per min; + Mt.Lion: 7.6 ± 1.0 per min. $N_{\text{mouse}} = 8$. RM one-way ANOVA: $F_{(2,14)} = 0.9016$, $p = 0.428$, ns). This behavior ~~also highlights that odorants emitted by food were sufficient to initiate this motivational-related behavioral process independently from GG detection (Ctrl, $N_{\text{mouse}} = 12$; Axo, $N_{\text{mouse}} = 12$; one-tailed w -test: $p = 0.483$, ns).~~ We then followed the systemic integration of these stressful situations with the expression of the immediate early gene c-Fos²¹. We focused on the amygdalopiriform transition area (APir; Fig. 3b) which is the brain region implicated in the increase of stress-related hormone level in the blood when mice smell volatile predator scents^{15,22}. We

found that this brain region was significantly activated by both SBT and the predator urine (Fig. 3c-d). We next confirmed this result by measuring an increase of the systemic corticosterone level (Fig. 3e), implying that intra- and inter-species danger cues were both processed in this specific APir nucleus. In a dilution series of intraperitoneal corticosterone injections (Cort. i.p.; Fig. 3f), we were further able to mimic the observed stress-related hormonal elevation with an amount of Cort. i.p. of 5.0 mg kg⁻¹, **therefore bypassing APir activation** in absence of conditioning stimuli (– C.S; Fig. 3f **and Supplementary Fig. 3**). Moreover, we found that, in Axo mice, the activation of the APir region as well as the elevation of the systemic corticosterone were significantly impaired in this threatening context (Fig. 3c to e), demonstrating that, although the GG was not involved **in the risk-taking of mice** in **odor-driven** foraging, it was essential for the hormonal and physiological adaptation to danger sensing.

Elevation of corticosterone optimizes learned-food selection. In the search of food resources, knowledge of familiar food obtained by STFP support the food decision-making by a recall memory process that requires the activation of a specific part of the hippocampus, the ventral subiculum^{13,23} (VS; Fig. 3b). In this study, we found that the presence of danger cues enhanced this learned food-**odorant** preference when we performed a two choices assay under threatening conditions. For that, **we first trained mice to develop a food-odorant preference. Briefly**, a demonstrator mouse **first** ate an odorized demonstrating food (de.food; standard powdered food odorized for example with Cinnamon as a spice #1; Phase 1; Fig. 4a). Then it returned with its observer littermates to allow the STFP learning process to happen (Phase 2; Fig. 4a). Finally, after 24 hours of fasting, the observer mice were individually confronted to a two choices assay between two odorized foods, the demonstrating food and a novel food (new.food; standard powdered food odorized for example with Cocoa as spice #2; Phase 3; Fig. 4b), both surrounded with the same conditioning stimulus (+ C.S; Fig. 4b). To avoid any individual innate preference, spices were used in a counterbalanced mode¹³ (Supplementary Fig. **4a** to c). We observed in both phenotypes that STFP was, indeed, required to develop a food preference (STFP+ | + Water; Fig. 4c) that was, remarkably, not impacted by a chemical confusion generated by the pungent BA (STFP+ | + BA; Fig. 4c). These first effects not only confirmed the conserved functionality of the GC-D circuitry in Axo mice (Fig. 1c **and e-f**) but also pointed out that, **under aversive distraction, mice still olfactively decipher**

food preferences. Surprisingly, we obtained, with the danger cue SBT (STFP+ | + SBT) and with Mt.Lion urine (STFP+ | + Mt.Lion) a significant enhancement of the food-odorant preference (Fig. 4c). Indeed, the observed mean food preferences were amplified by around 60 % compared to STFP under standard environmental (STFP+ | + Water) or under pungent conditions (STFP+ | + BA). This enhancement was observed in Ctrl but not in Axo mice. It was found to be both independent from food consumption (Supplementary Fig. 4 5a) and from an improvement of the recall memory process as no significant increase in c-Fos activity was observed in the VS brain area under these conditions (Fig. 4d and e). Interestingly, we found that injection of Cort. i.p. 5.0 mg kg⁻¹ under neutral context (STFP+ | Cort. i.p.) was sufficient to mimic this behavioral improvement both in Ctrl and Axo mice (Fig. 4c) **without affecting the memory retrieval activity of the VS (Supplementary Fig. 5b and c)**. We thus demonstrated here that, when danger cues are sensed by the GG, the increase in the systemic corticosterone level that occurs (Fig. 3e) leads to an enhancement of the food preference previously acquired by STFP.

GG circuitry activation selectively erase safety food memory. Conspecific interactions are useful for mice to get food familiarity. We found here that mice can also benefit from threatening chemical information present in their environment to reset this previously acquired food-odorant familiarity when associated with a danger context. We indeed tested the impact of olfactory threats on the acquisition of a **new food-odorant preference and thus as we** challenged mice 1 hour after an STFP procedure (Phase 1; Fig. 5a) with a surrogate display that contained the unreachable demonstrating food moistened with an associated conditioning stimulus (Phase 2; + C.S; Fig. 5a). After 24 hours of fasting, the observer mice were then tested in a two choices assay (Phase 3; Fig. 5b). Unexpectedly, we found that, in association with the danger cues SBT (STFP+ / + SBT) or with Mt.Lion urine (STFP+ / + Mt.Lion), Ctrl mice did not display food-odorant preferences while they were still observed in Axo mice (Fig. 5c). As a control, we verified that GG-related ligands directly associated with a demonstrating food could not act as conditioning stimuli promoting a food preference or avoidance (Supplementary Fig. 5 6a to c), confirming our previous observations that avoidance towards an unfamiliar odorized food is innately coded (Fig. 2d). Besides, we also observed that this apparent and selective amnesia was independent from food consumption (Supplementary Fig. 6 7a) or corticosterone elevation. Indeed, mice injected with Cort. i.p. 5.0 mg kg⁻¹

instead of the associated conditioning stimuli (STFP+ / Cort. i.p.) still displayed a food-odorant preference (Fig. 5c). Moreover, we observed a striking resetting of the VS brain area (c-Fos stainings; Fig. 5d and e), that strongly suggests an impairment in the recall memory for a food preference or for its consolidation. Remarkably, as this process was independent from the systemic corticosterone level (Supplementary Fig. 7b and c) and not observed in Axo mice, this cerebral reorientation adaptation was thus directly related with the activation of the GG neuronal circuitry itself (Fig. 5d and e).

Discussion

Acute stress can modulate both the performance of action-outcomes and the learning processes in animals including rodents and humans^{24,25}. Impending danger such as the risk of predation increases stress-related hormonal levels in the blood^{5,15} and can affect memorization²⁶. When faced with imminent danger, alerting senses like the olfactory senses allow the detection of chemical cues signaling for the presence of predators²⁷. The olfactory senses are also essential to find food and therefore trade-offs and sorting out of the environmental chemical information has to take place followed by the appropriate behavioral responses. In this study, we investigated the biological relevance of the detection of chemical danger cues by the GG on odor-driven food selection because of its fundamental need for animal survival. We demonstrated that the GG circuitry favors innate odor-driven food decision-making in a context of danger (Fig. 2). Indeed, the preys with a functional GG are able to select the food resource, depending on its smell, while avoiding to be eaten by predators. We have also observed that the GG was not directly implicated in the detection of odorants emitted by food resources (Fig. 2) and in the linked foraging behavior (Fig. 3), we can thus consider that our GG-dependent results are not exclusively specific to food selection^{15,17} and that they may be extended to the global effects of threatening scents on odor-driven preferences.

Our understanding of the interplay between the pathways processing GG-induced fear-like responses and those encoding food-odorant information remains to be completed. GG neurons use multi-signaling pathways, principally linked to bitter taste signaling²⁸ and guanylyl cyclase-G cascades¹⁸ to detect a large repertoire of chemical danger cues^{28,29}. Interestingly, in GC-D neurons, a similar encoding strategy occurs¹⁴ making it also sensitive to fatty acids, steroid hormones as well as to some GG-related

cues such as pyrazines. This shared encoding strategy as well as the partial overlap of recognized cues between the GC-D and the GG subsystems might, at the detection level, contribute to the apparent positive collaboration between these two parallel circuitries.

In a pilot experiment performed in the NG complex of GCG–Cre–GFP mice, we noticed the presence of periglomerular cells expressing the tyrosine hydroxylase^{30,31} (TH; Supplementary Fig. 8a and b). Interestingly, their density was preferentially located around GC-D glomeruli (GC-D: 9.3 ± 0.9 TH+ cells, $N_{glomeruli} = 6$; GG: 5.0 ± 1.2 TH+ cells, $N_{glomeruli} = 7$), supporting the interconnectivity of GC-D glomeruli with other olfactory circuitries such as those of the MOE³². On the other hand and compared to other OB regions³³, TH+ cell density was moderate around GG glomeruli, which is consistent with their homogenous innervation¹⁹ and their episodic afferent activity⁹. This might question the potential cross-glomerular regulatory task of these periglomerular cells. Moreover, and as a temporal hiatus, we found that the GG circuitry could impact socially learned food preferences in a delayed manner (Fig. 4 and Fig.5), at least 1 hour after the STFP procedure (Fig. 4). Therefore periglomerular regulation emerging from GG on GC-D glomeruli requires further investigations.

The integration of the GG circuitry on higher structures in the CNS have revealed specific subregions implicated in fear and anxiety responses such as the posteroventral division of the medial amygdala (MeA) and the dorsomedial subdivision of the ventromedial hypothalamus^{16,34} (VMH). Our findings unravel two additional brain domains implicated in this global integration, the APir (Fig. 3) and the VS (Fig. 4 and Fig. 5). Interestingly, the APir was previously reported as a specific area of the olfactory cortex that induces stress hormone responses to volatile predator scents²¹. Here, we demonstrate that this heterospecific response could be extended to conspecific-signaling integration, as the mouse alarm pheromone SBT likewise initiates APir activities (Fig. 3). In addition to the instinctive fear response, we therefore uncovered here a key consequence of the subsequent systemic corticosterone elevation and demonstrated its central outcome for the improvement of odor-driven food behavior (Fig. 4). The processing ability of the CNS structures, such as the one observed here with the APir is usually dependent of their activation. On the other hand, we found that the treatment of the information by the VS emerging from the GG circuitry consisted in its ability to reset its memory-retrieval activity for socially-learned food response^{13,23} (Fig. 5). This last finding is consistent with previous reports implying

threatening scents in the impairment of retrieval memory³⁵ but also supports the VS as a key CNS area implicated in context-dependent stress and integration of fear memories³⁶⁻³⁸. Moreover, we have observed that the same chemical danger cues can act both during (Fig. 4) and upstream (Fig. 5) of the decision-making event with opposing behavioral outcomes. Threatening scents could thus develop contrasting effects on odor-driven food choices in which the VS should play a context-dependent integrative role. These observations further imply a complex strategy for encoding both the predator-related stress as well as the retrieval of innate and learned fear³⁹. At the individual level, this innate trait could be considered as a beneficial mechanism of protection. Indeed, when the food choice is presented in a context of threat (Fig. 4), the favored decision for survival should be to select a safe and familiar food. Nonetheless, to be efficient and to decrease the risk of being eaten, a compromise is made and the emerging strategy is, in the end, to erase this food familiarity and to consume the food anyway (Fig. 3 and Fig. 5). Thus, for mice, as for humans, hierarchy of prepotency as well as behavioral motivation is innately coded. Therefore, physiological needs, such as feeding, surpass safety needs⁴⁰.

In the wild, the ability to take rapid decisions concerning food choice is critical for the survival of the individual and leads to an increase in the overall fitness of the species. It is therefore conserved throughout the animal kingdom^{41,42}. For example, in the nematode *Caenorhabditis elegans* (*C. elegans*), stress linked to environmental conditions can impact food preferences when they are coupled with odorants⁴³ as well as with alarm pheromones and predator scents^{44,45}. Interestingly, this worm modality is dependent on a specific olfactory subsystem composed of a pair of amphid wing cells of type C⁴⁶ (AWC). GG neurons were reported to share molecular and functional features with these AWC neurons, conferring to these two olfactory subsystems an inter-species orthologous status²⁹.

In summary, our present study demonstrates that the GG potentiates, in mice, the decision-making process for the odor-driven food choice under threat. Our results highlight how mice integrate environmental stressful stimuli to control both innate and previously acquired odor-driven food preferences. Olfactory threats are detected by the GG that operates as an immediate sensor acting both on innate perceptions, systemic hormonal levels and on the acquired food preferences to circumstantially revise mice decision-making, a mechanism that confers to the animal a beneficial survival advantage²¹.

Reviewers' comments:

Reviewer #1 (Remarks to the Author):

Since the authors re-organized the manuscript as I and other reviewer suggested, the manuscript has been greatly improved. However, there are still some concerns that have to be addressed.

1) There are concerns about statistics. The authors use a t-test or Wilcoxon test to investigate statistical significance even when they compare multiple data. I wonder if they performed correction for multiple comparisons. Besides, they should use two-tailed tests instead of one-tailed tests, because one cannot expect whether Axo mice show decreases or increases of preference or c-fos expression.

2) Throughout the text, the authors use "both phenotype" to indicate sham-operated and Axo mice. I am afraid that the use of "phenotype" could confuse readers because "phenotype" should be used when animals show some changes by genetic modifications. Therefore, the authors should not use "phenotype" but use other words, such as "Ctrl and Axo mice".

3) On page 9, the authors described that risk assessment behavior is not affected by GG axotomy. I recommend that this data is included in the main figures. In the previous paper (Brechtbühl 2013), the authors show that not risk assessment behavior but freezing behavior to predator odors are affected by GG axotomy. I wonder if freezing behavior is observed during the STFP experiment.

Reviewer #2 (Remarks to the Author):

Authors have carefully revised all my six major concerns and several typo corrections.

Reviewer #3 (Remarks to the Author):

The revised version has been massively re-written, and in my opinion much improved in terms of readability, removing fancy jargon and not overinterpreting the results. Now the original strength of the work is very visible. The authors have done a very good job in listening to the reviewers and implementing their suggestions/addressing their criticisms. All my concerns have been adequately addressed, and I recommend acceptance of the revised manuscript strongly.

Responses to reviewers:

COMMSBIO-20-0343A (The Grueneberg ganglion controls odor-driven food choices under threat).

Reviewer #1:

We would like to thank the reviewer for her/his comments as well as for her/his in-depth review of our revised manuscript. We have now addressed all of her/his concerns and recommendations. We are sure that taking them into account significantly strengthen our new revised manuscript.

In summary, we have now followed the advice of the reviewer and analyzed the comparisons between Ctrl and Axo mice and between conditions using the proposed two-tailed tests. We have now changed throughout the text the general terminology “phenotype” to the requested and more specific description “Ctrl and Axo mice”. We have now included data concerning risk assessment and freezing behaviors in the main text and in our new figure 3.

We hope that these modifications will allow a better understanding and interpretation of our results and we look forward to the reviewer’s response.

Please find here the responses to her/his specific comments:

Since the authors re-organized the manuscript as I and other reviewer suggested, the manuscript has been greatly improved. However, there are still some concerns that have to be addressed.

1) There are concerns about statistics. The authors use a t-test or Wilcoxon test to investigate statistical significance even when they compare multiple data. I wonder if they performed correction for multiple comparisons. Besides, they should use two-tailed tests instead of one-tailed tests, because one cannot expect whether Axo mice show decreases or increases of preference or c-fos expression.

Answer: We would like to thank the reviewer for raising these concerns. We would like to mention that no correction was applied to compare multiple data. As requested by the reviewer, we have now used two-tailed *t/w*-tests as no expectations concerning decreases or increases of preference or c-fos expression between Ctrl and Axo mice could indeed be made. Accordingly, we have now computed our statistical calculations using the two-tailed *t*-test or *w*-test. We have now modified the statistical description in the methods section and figure legends and adapted significance symbols in the figures.

We would like to mention that the use of these two-tailed *t/w*-tests does not modify the main message of our manuscript and its conclusion. When necessary, we now carefully revised our data description in the main text accordingly:

Main text, (line 83-85): “... independently from the GG axotomy procedure, as verified by immediate early gene *c-Fos* expression¹⁵ (Fig. 1e and f; Ctrl, $N_{glomeruli} = 6$; Axo, $N_{glomeruli} = 7$; ~~one~~ two-tailed *t*-test: $p = 0.272$ 0.544 , ns)”.

Main text, (line 127-128): “Nevertheless, in Axo mice, this dislike was *significantly* reduced...”.

Main text, (line 196-198): “Surprisingly, we obtained, with the danger cue SBT (STFP+ | + SBT) and with Mt.Lion urine (STFP+ | + Mt.Lion) an *significant* enhancement of the food-odorant preference (Fig. 4c).”.

2) Throughout the text, the authors use “both phenotype” to indicate sham-operated and Axo mice. I am afraid that the use of “phenotype” could confuse readers because “phenotype” should be used when animals show some changes by genetic modifications. Therefore, the authors should not use “phenotype” but use other words, such as “Ctrl and Axo mice”.

Answer: We appreciate this remark. We have now systematically changed this terminology. For example, please find below representative modifications:

Main text, (line 137): “..., **Ctrl and Axo** mice ~~from both phenotypes~~ now preferred Cinnamon, ...”.

Main text, (line 159): “in both ~~phenotypes~~ **Ctrl and Axo mice**...”.

Legend Fig. 3, (line 732-733): “For comparisons between condition and ~~phenotypes~~ **between Ctrl vs. Axo mice**, ...”.

3) On page 9, the authors described that risk assessment behavior is not affected by GG axotomy. I recommend that this data is included in the main figures. In the previous paper (Brechtbühl 2013), the authors show that not risk assessment behavior but freezing behavior to predator odors are affected by GG axotomy. I wonder if freezing behavior is observed during the STFP experiment.

Answer: We would like to thank the reviewer for this fundamental comment. We agree with her/him on the importance of these behaviors and their potential subsequent influence on the STFP procedure. Indeed, as mentioned by the reviewer, these behaviors could take place during the STFP experiment described on page 9 (Fig. 5a, Phase 2) similarly to the situation described in our Fig. 3a (we have now mentioned this point in the main text, *line 217-218*). We have now extracted from our risk assessment data (Fig. 3a), implemented with new behavioral sessions, freezing responses of mice in this context. We were surprised to observe that, in addition to developing risk assessment behavior, fasting mice did not develop freezing response when confronted to search for food resource, even in the presence of a threat (new Fig. 3b). We believe that this new observation strongly supports our initial observations and interpretations regarding trade-off displayed by fasting mice. Thus, in a scarceness context, the opportunity to eat indeed overrules fear. Food-related odorants are indeed sufficient to initiate motivational-related behavioral process. According to the reviewer recommendation, we have now included and

centralized data concerning the observed mice risk assessment and freezing responses that could occur during active food research and the STFP experiment in our new figure 3 (new Fig. 3b-c) and we have adapted the main text accordingly.

Original Figure 3

Revised Figure 3

Responses to reviewers:

COMMSBIO-20-0343A (The Grueneberg ganglion controls odor-driven food choices under threat).

Reviewer #2:

We would like to thank the reviewer for her/his contribution to our work and for accepting our revised manuscript.

We would like to mention that we have now addressed the concerns and recommendations of reviewer #1 in a newly revised manuscript. In summary, we have calculated the statistics accordingly for comparisons between Ctrl and Axo mice and between conditions. We have now changed throughout the text the general terminology “phenotype” to the requested and more specific description “Ctrl and Axo mice”. We have now included data concerning risk assessment and freezing behaviors in the main text and in our new figure 3.

Responses to reviewers:

COMMSBIO-20-0343A (The Grueneberg ganglion controls odor-driven food choices under threat).

Reviewer #3:

We would like to thank the reviewer for her/his contribution to our work and for accepting our revised manuscript.

We would like to mention that we have now addressed the concerns and recommendations of reviewer #1 in a newly revised manuscript. In summary, we have calculated the statistics accordingly for comparisons between Ctrl and Axo mice and between conditions. We have now changed throughout the text the general terminology “phenotype” to the requested and more specific description “Ctrl and Axo mice”. We have now included data concerning risk assessment and freezing behaviors in the main text and in our new figure 3.

REVIEWERS' COMMENTS:

Reviewer #1 (Remarks to the Author):

The authors have revised their manuscript that satisfactorily addressed most of my concerns, except for one concerning statistics with multiple comparisons. Without corrections to compare multiple data, the possibility of the type I error is increased. The authors should at least declare that they applied no correction to compare multiple data in the method section.

Responses to reviewers:

COMMSBIO-20-0343B (The Grueneberg ganglion controls odor-driven food choices under threat).

Reviewer #1:

We would like to thank the reviewer for her/his comments as well for her/his substantial contribution to our work. We have now addressed her/his specific comment (please find our detailed answer on the next page) and, as recommended by the reviewer, **we have now explicitly stated in the method section that we applied no correction to compare multiple data.**

We hope that this clarification will allow a better appreciation and interpretation of our results.

Please find here the responses to her/his specific comments:

The authors have revised their manuscript that satisfactorily addressed most of my concerns, except for one concerning statistics with multiple comparisons. Without corrections to compare multiple data, the possibility of the type I error is increased. The authors should at least declare that they applied no correction to compare multiple data in the method section.

Answer: We would like to thank the reviewer for raising this concern and apologize for the lack of clarity in our previous response. Here we would like to take the opportunity to clarify our statistical strategy regarding multiple comparisons.

We agree with the reviewer that multiple comparisons indeed increase the possibility of false positive results (error of type I) and thus the emergence of non-specific significant results. On the other hand, corrections applied to multiple comparisons increase the possibility of false negative results (error of type II). Therefore, we initially opted to focus on a subset of relevant comparisons (indicated by connecting lines in figures) by using *t/w*-tests and applying their two-tailed procedure, as previously suggested by the reviewer. Furthermore, we have included in our experiments numerous controls such as the “STFP–” condition, the odorless “Water”, the aversive and non-threatening “Butyric acid” (BA) and related candidates such as “Corticosterone” (Cort. i.p.) that are supporting the same conclusion: *the impact of threatening scents on food choices*. We have thus considered that the inclusion of these control conditions in the same experimental assay were internally consistent and did not require to compensate for multiple comparisons (D.S. Fay, K. Gerow, Wormbook, 2013).

In the case of multiple and non-focused comparisons performed (please see the total food consumption in our supplementary data), we think that the reviewer’s concern about the potential false positive results is justified. Nevertheless, the potential emergence of these type I errors was not observed in our case as only non-significant differences were obtained between the conditions tested which confirms, in our opinion, the consistency of our results and of our conclusions.

To avoid any confusion and as recommended by the reviewer, **we have now explicitly stated in the method section that we applied no corrections to compare multiple data.**